# SurvDiff: A Diffusion Model for Generating Synthetic Data in Survival Analysis

## Abstract

Survival analysis is a cornerstone of clinical research by modeling time-to-event outcomes such as metastasis, disease relapse, or patient death. Unlike standard tabular data, survival data often come with incomplete event information due to dropout, or loss to follow-up. This poses unique challenges for synthetic data generation, where it is crucial for clinical research to faithfully reproduce both the event-time distribution and the censoring mechanism. In this paper, we propose SurvDiff, an *end-to-end diffusion model specifically designed for generating synthetic data in survival analysis*. SurvDiff is tailored to capture the data-generating mechanism by jointly generating mixed-type covariates, event times, and right-censoring, guided by a survival-tailored loss function. The loss encodes the time-to-event structure and directly optimizes for downstream survival tasks, which ensures that SurvDiff (i) reproduces realistic event-time distributions and (ii) preserves the censoring mechanism. Across multiple datasets, we show that SurvDiff consistently outperforms state-of-the-art generative baselines in both distributional fidelity and survival model evaluation metrics across multiple medical datasets. To the best of our knowledge, SurvDiff is the first diffusion model explicitly designed for generating synthetic survival data

## 1 Introduction

Survival analysis is a core tool in medicine for modeling time-to-event outcomes (the duration until an event occurs), such as progression-free survival in cancer or overall survival in clinical trials (Bewick et al., 2004; Arsene & Lisboa, 2007). Unlike standard tabular datasets, survival data are characterized by *right-censoring*,

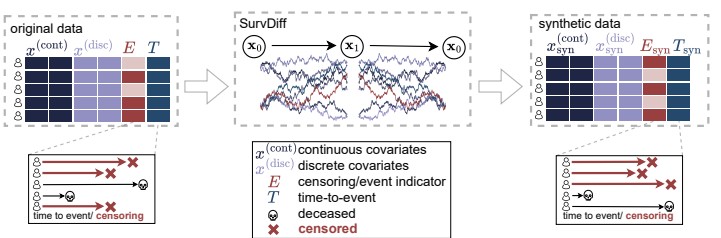

Figure 1: **SurvDiff for generating synthetic survival data.** Our SurvDiff generates synthetic samples that retain the structure of the original data, including high-fidelity covariate distributions and faithful event-time distributions while preserving the *censoring mechanism*. The synthetic dataset can then be used to train downstream survival models without direct access to the original patient-level data.

*censoring*, where events are not observed due to dropout, loss to follow-up, or adverse reactions. Such right-censoring is common in medical practice and can affect nearly half of patients in some cancer trials (Shand et al., 2024; Norcliffe et al., 2023).

However, generating synthetic data for survival analysis is particularly *challenging* because failing to correctly model censoring mechanisms can bias downstream clinical results (Norcliffe et al., 2023; Wiegrebe et al., 2024). Unlike standard tabular data generation, the task requires not only capturing covariate distributions but also faithfully (i) *reproducing time-to-event distributions* and (ii) *preserving censoring mechanisms* (Bender et al., 2021). This interplay between covariates, survival times, and censoring makes survival data generation inherently more complex than standard tabular synthesis and is why naïve applications of generic synthetic data methods, such as standard generative adversarial networks (GANs) or diffusion models, fail in survival contexts.

To the best of our knowledge, there exist only two methods tailored method for generating synthetic survival data (see Table 1): SurvivalGAN (Norcliffe et al., 2023) and the framework of Ashhad and

Henao (Ashhad & Henao, 2024; 2025) (which we refer to as *Ashhad* in the following). Both Sur-vivalGAN and Ashhad decompose survival data generation into separate components for covariates and for event times and censoring, rather than learning a single joint model. However, these ap-proaches have major **limitations**: **(1)** in the case of SurvivalGAN, the GAN backbone is prone to mode collapse and therefore unstable training; **(2)** they rely on multi-stage pipelines with different models for covariates and event-time mechanisms, which makes them prone to error propagation and prevents end-to-end learning. As a result, SurvivalGAN and Ashhad produce distributions of covariates, event times, and censoring of limited fidelity.

Recently, diffusion models (Sohl-Dickstein et al., 2015; Shi et al., 2024b; Zhang et al., 2024) have gained popularity as a powerful tool for generating synthetic *tabular* data. Diffusion models of-fer stable training, avoid mode collapse, and consistently achieve high fidelity across diverse do-mains (Dhariwal & Nichol, 2021; Chen et al., 2024), which makes them a strong candidate for our task. However, they are *not* designed for survival data, and, as we show later, a naïve application thus fails to (i) reproduce realistic event-time distributions and (ii) preserve censoring mechanisms. To the best of our knowledge, a diffusion model tailored specifically to generating synthetic survival data is still missing.

In this paper, we propose SURVDIFF, a novel *end-to-end diffusion model for generating synthetic survival data*. Our SURVDIFF is carefully designed to address the unique challenges of survival data. For this, SURVDIFF *jointly* generates covariates, event times, and right-censoring, guided by a survival-tailored loss function. Our novel loss encodes the time-to-event structure and explicitly accounts for censoring, ensuring that SURVDIFF (i) reproduces realistic event-time distributions and (ii) preserves censoring mechanisms. We further improve training stability with a *sparsity-aware weighting scheme* that accounts for right-censoring by giving higher weight to earlier event times, which have more support in the data, and lower weight to later event times, which have less support. Together, these design choices allow SURVDIFF to generate synthetic survival datasets that are faithful regarding both covariate distributions and survival outcomes.

Our **main contributions**[1] are the following: **(1)** We propose a novel, diffusion-based method called SURVDIFF for synthetic data generation in survival settings. **(2)** Unlike existing methods, our SURVDIFF is end-to-end, which allows it to *jointly* optimize covariate fidelity and time-to-event information under censoring. **(3)** We conduct extensive experiments across multiple datasets from medicine, where we demonstrate that our SURVDIFF achieves state-of-the-art performance in both producing high-fidelity data and downstream survival analysis. In particular, we show that our SURVDIFF outperforms naïve applications of tabular diffusion models in ablation studies.

## 2    RELATED WORK

Generating synthetic data is often relevant for several reasons, such as augmenting datasets (Perez & Wang, 2017), mitigating bias and improving fairness (van Breugel et al., 2021), and promoting data accessibility in low-resource healthcare settings (de Benedetti et al., 2020). While synthetic data is widely explored for images and medical domains (Amad et al., 2025), less attention has been given to survival data (see below).

**ML for survival analysis:** Machine learning for survival analysis faces unique challenges (Wiegrebe et al., 2024; Frauen et al., 2025) because survival data combine time-to-event outcomes with right-censoring, which makes standard supervised learning methods inapplicable.

Traditional statistical approaches estimate hazard ratios or survival curves (Bender et al., 2005; Austin, 2012). More recently, deep learning methods have adapted to this setting (Ranganath et al., 2016; Miscouridou et al., 2018; Zhou et al., 2022) but often with restrictive parametric assumptions (e.g., Weibull distribution), or with conditioning on covariates (Bender et al., 2021; Kopper et al., 2022). Importantly, the focus is on estimating survival times, but *not* generating complete synthetic datasets including covariates, event times, and censoring information (Konstantinov et al., 2024).

**Synthetic data generation for *tabular* data:** A range of generative models has been proposed for generating synthetic tabular data (see overview in Shi et al. (2025)). These are often based on normalizing flows (**NFlow**) (Papamakarios et al., 2021), variational autoencoders (VAE) (Kingma & Welling, 2013), and generate adversarial networks (GAN) (Goodfellow et al., 2014). Further, several

---

[1]Code is available at `https://anonymous.4open.science/r/SurvDiff-E6A0`. Upon accep-tance, we move our code to a public GitHub repository.

specialized versions have been developed, such as: **CTGAN** (Xu et al., 2019) extends the GAN framework to mixed-type covariates using mode-specific normalization and conditional sampling. **TVAE** (Xu et al., 2019) leverages variational autoencoders to encode and recreate heterogeneous feature types. However, these methods are *not* reliable in avoiding instability or mode collapse during training (Saxena & Cao, 2021; Gong et al., 2024).

More recent work has turned to ***diffusion models*** (Sohl-Dickstein et al., 2015; Song & Ermon, 2019; Ho et al., 2020; Song et al., 2021), which recently emerged as a powerful alternative for tabular data generation and which offers improved stability and fidelity compared to adversarial or variational methods. A state-of-the-art method here is **TabDiff** (Shi et al., 2024b), which directly builds on the earlier TabDDPM model for tabular data (Kotelnikov et al., 2023). As such, diffusion models established strong baselines for synthetic tabular data and remain widely used. However, these methods remain general-purpose and are *not* designed to (i) handle time-to-event outcomes or (ii) preserve censoring. Still, we later use the above state-of-the-art tabular diffusion model as a baseline.

**Synthetic data generation for *survival* data:** To the best of our knowledge, there are only two tailored for survival data generation, namely, **SurvivalGAN** (Norcliffe et al., 2023) and the **Ashhad** framework (Ashhad & Henao, 2024; 2025). Both methods generate the factorized distribution in stages rather than jointly. While these approaches demonstrate the feasibility of generating synthetic survival data, **(1)** in case of SurvivalGAN, the GAN backbone is prone to mode collapse and unstable training; and **(2)** for both methods, the staged design and reliance on multiple components make it more prone to error propagation.

**Research gap:** To the best of our knowledge, there is no tailored diffusion model for generating synthetic survival data (Table 1). To fill this gap, we propose SURVDIFF, which is the first end-to-end diffusion model for that purpose and which addresses key limitations of existing baselines.

| Datatype | Model | Backbone | Survival[†] | **Key generative models for synthetic data generation in our context.** | | |
|---|---|---|---|---|---|---|
| **Tabular** | NFlow | Flows | ✗ | While there is a large stream of generative models for tabular data, methods tailored to survival data (e.g., preserving censoring mechanisms) are scarce. | | |
| | TVAE | VAE | ✗ | | | |
| | CTGAN | GAN | ✗ | | | |
| | TabDiff | Diffusion | ✗ | | | |
| | | | | **High-fidelity patient covariates** | **End-to-end** | **Avoid error propagation** |
| **Survival** | SurvivalGAN | GAN | ✓ | ✓ | ✗ | ✗ |
| | Ashhad | model-agnostic | ✓ | ✓ | ✗ | ✗ |
| | **SURVDIFF (*ours*)** | **Diffusion** | ✓ | ✓ | ✓ | ✓ |

[†] Survival data generation models tailored to time-to-event and censoring.

Table 1: Key works on synthetic data generation.

# 3 SETTING

**Notation.** We denote random variables by capital letters $X$ and realizations by small letters $x$. We write the probability distribution over $X$ as $P_X$ and as $p(x)$ its probability mass function for discrete variables or the probability density function w.r.t. the Lebesgue measure for continuous variables.

## 3.1 MATHEMATICAL BACKGROUND

**Diffusion models:** Diffusion models (Sohl-Dickstein et al., 2015; Song & Ermon, 2019; Ho et al., 2020; Song et al., 2021) define a generative process by perturbing data through a forward noising scheme and then learning a reverse procedure. (1) The *forward process* begins from data samples $x_0 \sim P_X$ and evolves according to a Markovian stochastic differential equation (SDE) indexed by a diffusion time $u \in [0, 1]$ via

$$\mathrm{d}x = f(x, u)\,\mathrm{d}u + g(u)\,\mathrm{d}w_u, \tag{1}$$

where $f$ is the drift term, $g$ the diffusion coefficient, and $w_u$ a Wiener process, i.e., a Brownian motion with independent Gaussian increments $W_{u+\Delta} - W_u \sim \mathcal{N}(0, \Delta I)$. As $u$ increases, the distribution $P_u$ converges to a tractable noise distribution, typically Gaussian. (2) By reversing the process, one can then sample from the original distribution. Under mild regularity conditions, the reverse-time dynamics satisfy

$$\mathrm{d}x = \left[ f(x, u) - g(u)^2 \nabla_x \log p_u(x) \right] \mathrm{d}u + g(u)\,\mathrm{d}\bar{w}_u, \tag{2}$$

where $\bar{w}_u$ is a reverse-time Wiener process and $\nabla_x \log p_u(x)$ the score function, i.e., the gradient of the log density at noise level $u$. Because the score function is unknown, a neural network $\mu_\theta(x, u)$ is trained via score-matching to approximate $\nabla_x \log p_u(x)$. Once trained, the model can approximate the reverse SDE and transform Gaussian noise into samples from the target distribution.

The above diffusion model provides a tractable approximation to maximum likelihood and underlies a broad family of generative models. However, in its standard form, it cannot model mixed-type variables (continuous or discrete), because of which extensions such as TabDiff (Shi et al., 2024b) are used. More importantly for our setting, while there are some extensions to medical settings (Ma et al., 2024; Amad et al., 2025; Ma et al., 2025), there is *no* diffusion model to capture the censoring mechanism in survival data, which motivates the need for a tailored method.

**Survival analysis:** The goal of survival analysis (Bewick et al., 2004; Machin et al., 2006) is to model the time until an event of interest (e.g., metastasis, relapse, etc.) occurs. For simplicity, we assume that death is the event of interest. In practice, the event is not always observed because of censoring. Let $T \geq 0$ denote the *censoring time* if the event is censored ($E = 0$), and the *event time* if the event was observed ($E = 1$).

The *survival function* $S(t \mid x) = p(T > t \mid X = x)$ for individuals with covariates $X = x$ at time $t$ that quantifies the probability of surviving beyond $t$ given covariates $x$. The event process can be equivalently expressed through the *hazard function* $h(t \mid x) = \lim_{\Delta t \to 0} \frac{p(t \leq T < t + \Delta t \mid T \geq t, X = x)}{\Delta t}$, which gives the instantaneous risk of death at time $t$ conditional on surviving up to $t$. Survival and hazard functions are linked via $S(t \mid x) = \exp\left(-\int_0^t h(s \mid x) \, \mathrm{d}s\right)$. The expected time-to-event is $\mathbb{E}[T \mid x] = \int_0^\infty S(t \mid x) \, \mathrm{d}t$ (or a finite-time horizon when the study horizon is restricted). In practice, the survival probabilities $S(t \mid x)$ are estimated from $(X_i, E_i, T_i)$ using tailored models for censored time-to-event data, for example, Cox proportional hazards regression (e.g., Cox, 1972), which parameterize either the hazard or the survival function while accounting for censoring.

### 3.2 PROBLEM STATEMENT

**Data:** We observe an i.i.d. dataset $\mathcal{D}_{\text{real}} = \{(x_i^{(\text{disc})}, x_i^{(\text{cont})}, E_i, T_i)\}_{i=1}^n$ with patient data drawn from some distribution $P$, which consists of (1) continuous covariates $x_i^{(\text{cont})} \in \mathbb{R}^{d_{\text{cont}}}$, (2) discrete covariates $x_i^{(\text{disc})} = \left(x_{i,1}^{(\text{disc})}, \ldots, x_{i,d_{\text{disc}}}^{(\text{disc})}\right) \in \mathbb{R}^{d_{\text{disc}}}$ with one-hot encoding, (3) the event indicator $E_i \in \{0, 1\}$, and (4) an observed event time $T_i \in \mathbb{R}_+$. Here, censoring is captured by the event indicator, which denotes whether the event was observed ($E_i = 1$) or whether it was censored ($E_i = 0$), such as due to study dropout, loss of follow-up, or adverse reactions.

**Task:** Given the original data $\mathcal{D}_{\text{real}}$, our objective is to generate $\tilde{n}$ new samples $\mathcal{D}_{\text{syn}} = \{(x_i^{(\text{disc})}, x_i^{(\text{cont})}, E_i, T_i)\}_{i=1}^{\tilde{n}}$ that approximate the target distribution $P$. In particular, the synthetic data $\mathcal{D}_{\text{syn}}$ must preserve both (i) the joint distribution of covariates and (ii) survival outcomes (i.e., the time-to-event information as induced by the censoring mechanism conditional on covariates).

**Fidelity desiderata.** As in previous literature (Norcliffe et al., 2023), we measure the closeness of $\mathcal{D}_{\text{syn}}$ to $\mathcal{D}_{\text{real}}$ along four main dimensions:

(i) *Covariate fidelity.* Here, the idea is to generate patient samples that have similar characteristics (e.g., age, gender, etc.) as the original dataset. Optimally, $\mathcal{D}_{\text{real}}$ and $\mathcal{D}_{\text{syn}}$ should be drawn from the same distribution $P$. This similarity can be quantified via distances such as the Jensen–Shannon distance or the Wasserstein distance.

(ii) *Survival-specific fidelity.* We assess whether the synthetic data $\mathcal{D}_{\text{syn}}$ capture the temporal structure of the survival process. This includes the Event-Time Divergence (ETD) metric, which compares covariates of individuals experiencing events in similar time intervals, and temporal distribution plots for censored and uncensored events.

(iii) *Overall fidelity.* To evaluate fidelity across all variables, we report the Shape metric (Shi et al., 2024b), which incorporates $T$ and $E$ and compares marginal distributions, and provide normalized marginal histograms for $X, T$, and $E$ to compare real and synthetic marginal distributions.

(iv) *Survival analysis performance.* The aim is to generate data that allow training survival models on synthetic samples and evaluating them on real outcomes. This follows the idea of *train on synthetic, test on real* (TSTR) to assess the ability of the synthetic data to be used for real-world applications (Esteban et al., 2017). In our case, we evaluate whether the synthetic data $\mathcal{D}_{\text{syn}}$ preserves event-time structure. We report the concordance index (Harrell et al., 1982) (C-index), which measures correct risk ranking, and the Brier score (Brier, 1950), which measures the accuracy of predicted survival probabilities.

Below, we develop a diffusion model tailored to survival data, yet where preserving censoring is non-trivial. Unlike standard diffusion models, our method incorporates a censoring-aware objective to generate synthetic data with event-time and censoring patterns that align with the real data $\mathcal{D}_{\text{real}}$.

## 4 METHOD

**Overview.** We now introduce SURVDIFF, a diffusion-based model for generating synthetic survival data in an *end-to-end* manner, where we jointly model both continuous and discrete covariates, event times, and censoring indicator. SURVDIFF comprises three components (see Figure 4.3): **A** a *forward diffusion process* that perturbs covariates, event times, and censoring indicators; **B** a *reverse diffusion process* that reconstructs survival data from noise; and **C** a *survival-tailored diffusion loss* that preserves event-time ordering while incorporating censored observations.

In SURVDIFF, we employ a masked-diffusion process (Sahoo et al., 2024) together with a Gaussian diffusion process, and follow the architecture in Shi et al. (2024b) to handle mixed-type covariates. The main novelty lies in how we design the training objective, which enables learning high-fidelity covariate distributions and thus generates faithful synthetic datasets for downstream survival tasks. We distinguish the role of the event indicator $E$ (discrete) and event time $T$ (continuous), which progress along different noising schemes due to the different variable types.

To integrate both continuous and discrete variables, we represent the continuous covariates jointly in a vector of dimension $d_{\text{cont}}$ and encode the discrete covariates each in a one-hot vector. Specifically, for individual $i$ and covariate $j$ with $C_j$ different values, we obtain $x_{i,j}^{(\text{disc})} \in \mathcal{V}_j = \{v \in \{0,1\}^{C_j+1} \mid \sum_{k=1}^{C_j+1} v_k = 1\}$, where the first $C_j$ entries correspond to the different values and the last entry to a mask state. The mask is later used to hide specific one-hot vectors, forcing the model to learn the original value of the discrete covariate. We denote the one-hot vector representing the mask by $m \in \mathcal{V}_j$ with $m_k = 1$. In addition, we define $P_{\text{cat}}(\cdot; \pi)$ as the discrete distribution over the $C_j$ possible values and the mask with probabilities $\pi \in \Delta^{C_j+1}$, where $\Delta^{C_j+1}$ is the $C_j + 1$-simplex. For simplification, with a slight abuse of notation, we omit the index $i$ for a patient in the following.

### 4.1 **A** FORWARD DIFFUSION PROCESS

Following Shi et al. (2024b), the forward diffusion process in SURVDIFF perturbs each element of the data point $(x^{(\text{cont})}, x^{(\text{disc})}, E, T)$ with the power-mean noise schedule $\sigma^{\text{cont}}(\cdot)$ and the log-linear noise schedule $\sigma^{\text{disc}}(\cdot)$ for continuous and discrete covariates. We review both cases below.

• *Continuous covariates:* Let $z = (x^{(\text{cont})}, T)$. We adopt a so-called variance-exploding (VE) SDE (Song et al., 2021; Karras et al., 2022; Shi et al., 2024b):

$$\mathrm{d}z = f(z, u)\,\mathrm{d}u + g(u)\,\mathrm{d}W_u, \qquad f(z, u) \equiv 0, \quad g(u) = \sqrt{\tfrac{\mathrm{d}}{\mathrm{d}u}(\sigma^{\text{cont}}(u))^2}. \tag{3}$$

where $W_u$ is a standard Wiener process. The forward perturbation then has the closed form

$$z_u = z_0 + \sigma^{\text{cont}}(u)\varepsilon, \quad \varepsilon \sim \mathcal{N}(0, I_{d_{\text{cont}}}), \quad q(z_u \mid z_0) = \mathcal{N}(z_0, (\sigma^{\text{cont}}(u))^2 I_{d_{\text{cont}}}), \tag{4}$$

with identity matrix $I$ and $z_0$ the embedding of the original data point $(x^{(\text{cont})}, T)$ with diffusion time $u = 0$. As $\sigma^{\text{cont}}(u)$ increases, the marginal distribution converges to isotropic Gaussian noise, while each conditional remains centered at the transformed $z_0$.

• *Discrete covariates:* Let $\tilde{z} = (x^{(disc)}, E)$ and $\tilde{z}_0$ the embedding of the original data point $(x^{(disc)}, E)$. We use a masking process (Austin et al., 2021; Shi et al., 2024a; Sahoo et al., 2024; Shi et al., 2024b) with schedule $\alpha_u = \sigma^{\text{disc}}(u) \in [0, 1]$, where $\alpha_u$ decreases monotonically in $u$. At each step, a one-hot vector representing a discrete value is retained with probability $\alpha_u$ and replaced by the mask $m$ with probability $1 - \alpha_u$ via

$$q(\tilde{z}_u \mid \tilde{z}_0) = p_{\text{cat}}(\tilde{z}_u; \alpha_u \tilde{z}_0 + (1 - \alpha_u)m). \tag{5}$$

As $u \to 1$, all entries converge to the mask state, such that the representation loses informative structure and becomes indistinguishable across samples.

### 4.2 **B** REVERSE DIFFUSION PROCESS

We now aim to model the underlying survival data distribution $P$. For this, the reverse process in SURVDIFF reconstructs survival data from noisy inputs by iteratively denoising the continuous and discrete covariates together with the event indicator and event time. The denoising network, parameterized by $\theta$, produces outputs for covariates and survival quantities. The diffusion loss $\mathcal{L}_{\text{diff}}$ guides training for feature reconstruction while the survival loss $\mathcal{L}_{\text{surv}}$ enforces time-event structure.

• *Continuous covariates:* The reverse-time VE dynamics are parameterized by the score function $\nabla_z \log p_u(z)$ with $z = (x^{(\text{cont})}, T)$, which transports samples from Gaussian noise back to valid

data points. To do so, we train a diffusion model $\mu_\theta$, with the continuous part of the model output $\mu_\theta^{\text{cont}}$, to predict the perturbation $\varepsilon$ in the closed-form $z_u = z_0 + \sigma^{\text{cont}}(u)\varepsilon$. Here, the objective is

$$\mathcal{L}_{\text{cont}}(\theta) = \mathbb{E}_{z_0 \sim P_{T, X^{(\text{cont})}}} \mathbb{E}_{u \sim U[0,1]} \mathbb{E}_{\varepsilon \sim \mathcal{N}(0, I_{d_{\text{cont}}})} \left[ \left\| \mu_\theta^{\text{cont}}(z_u, u) - \varepsilon \right\|_2^2 \right], \tag{6}$$

which is equivalent (up to weightings) to score matching for VE SDEs. The diffusion model $\mu_\theta$, with the continuous part of the model output $\mu_\theta^{\text{cont}}$, reconstructs the original datapoints $z_0$ from the noisy data.

• *Discrete covariates:* For $\tilde{z} = (x^{(\text{disc})}, E)$ with masking schedule $\alpha_u = \sigma^{\text{disc}}(u)$, the reverse dynamics progressively denoise the original values from the mask $m$. The distribution of $\tilde{z}$ over an earlier index $s < u$ is given by

$$q(\tilde{z}_s \mid \tilde{z}_u, \tilde{z}_0) = \begin{cases} p_{\text{cat}}(\tilde{z}_s; \tilde{z}_u), & \tilde{z}_u \neq m, \\ p_{\text{cat}}\left(\tilde{z}_s; \frac{\alpha_s - \alpha_u}{1 - \alpha_u} \tilde{z}_0 + \frac{1 - \alpha_s}{1 - \alpha_u} m\right), & \tilde{z}_u = m. \end{cases} \tag{7}$$

The diffusion model $\mu_\theta$, with the discrete part of the model output $\mu_\theta^{\text{disc}}$, reconstructs the original datapoint $\tilde{z}_0$ from the noisy inputs. The objective follows from the continuous-time evidence lower bound (ELBO) for masking diffusion

$$\mathcal{L}_{\text{disc}}(\theta) = \mathbb{E}_{\tilde{z}_0 \sim P_{E, X^{(\text{disc})}}} \left[ \int_0^1 \frac{\dot{\alpha}_u}{1 - \alpha_u} \log \langle \mu_\theta^{\text{disc}}(\tilde{z}_u, u), \tilde{z}_0 \rangle \, \mathbf{1}[\tilde{z}_u = m] \, \mathrm{d}u \right]. \tag{8}$$

with $\dot{\alpha}_u = \frac{\mathrm{d}}{\mathrm{d}u} \alpha_u$ and where $\langle \cdot, \cdot \rangle$ is the inner product.

**Diffusion loss:** The overall **diffusion loss** is obtained as a weighted combination of continuous and discrete terms with weights $\lambda_{\text{cont}}, \lambda_{\text{disc}} > 0$:

$$\mathcal{L}_{\text{diff}}(\theta) = \lambda_{\text{cont}} \mathcal{L}_{\text{cont}}(\theta) + \lambda_{\text{disc}} \mathcal{L}_{\text{disc}}(\theta). \tag{9}$$

### 4.3 ⓒ SURVIVAL-TAILORED DIFFUSION LOSS

To encode the survival-specific data structure, including event times and censoring indicator, SURVDIFF adds a survival loss on top of the diffusion objective. Concretely, we generate a prediction of survival risk from the denoised covariates and adapt the loss to account for regions with uneven data support, thereby ensuring that rare long-term events are not overweighted.

Let $x^{(\text{cont})} \in \mathbb{R}^{d_{\text{cont}}}$ denote the predicted continuous vector, and let $x_j^{(\text{disc})} \in \mathcal{V}_j$ be the predicted probability vector for discrete covariate $j$ (including the [mask] state). We concatenate these to form $\mathbf{x} = [x^{(\text{cont})}; x_1^{(\text{disc})}; \ldots; x_{d_{\text{disc}}}^{(\text{disc})}]$. A survival head $f_\theta$, realized as a multi-layer perceptron, maps $\mathbf{x}$ to a scalar risk score $r = f_\theta(\mathbf{x})$. Now, consider sample $i = 1, \ldots, n$ with observed times $T_i$, event indicators $E_i$, and risk sets $\mathcal{R}(T_i) = \{k \in [n] : T_k \geq T_i\}$. The risk set at time $T_i$ contains all patients who are still under observation and have not yet experienced the event.

Our survival loss extends the Cox partial negative log-likelihood (Cox, 1972; Katzman et al., 2018) with **sparsity-aware weighting**, which models the event risk proportional to a baseline hazard and covariate effects over time. We optimize

$$\mathcal{L}_{\text{surv}}(\theta) = - \sum_{i \in [n]: E_i = 1} w_i \log \frac{\exp(r_i)}{\sum_{j \in \mathcal{R}(T_i)} \exp(r_j)}, \tag{10}$$

with the predicted scalar risk score $r_i$ and the importance weights $w_i$ defined below to balance the contributions across event times and mitigate sparsity in regions with limited support. Only uncensored events ($E_i = 1$) contribute directly; censored observations affect the denominator via the risk sets. With $w = 1$, our loss simplifies to the classical Cox proportional hazards loss (Katzman et al., 2018).

In our loss, we choose $w_i$ as follows. First, we note that late events yield small risk sets and unstable gradients. Hence, our $w_i$ should downweight rare long-duration events while preserving the partial-likelihood structure. For event $i$ within time $T_i$, we define

$$w_i = \begin{cases} 1, & T_i \leq \tau, \\ \exp\left(-\alpha (T_i - \tau)\right), & T_i > \tau, \end{cases} \tag{11}$$

where $\tau$ is the duration threshold (e.g., 80th percentile of the maximum observed time) from which exponential downweighting starts. Therein, we use an exponential decay weighting to downweight rare late events, which reduces instability from small risk sets and makes the joint optimization of diffusion and survival objectives more stable, while remaining differentiable.

**Overall SURVDIFF loss:** Then, the total loss consisting of the multiple objectives is

$$\mathcal{L}_{\text{total}}(\theta) = \mathcal{L}_{\text{diff}}(\theta) + \lambda_{\text{surv}}\mathcal{L}_{\text{surv}}(\theta) \tag{12}$$

with $\lambda_{\text{surv}} > 0$ and initiated adaptively. This formulation allows SURVDIFF to be trained end-to-end, jointly aligning feature reconstruction with survival-specific objectives.

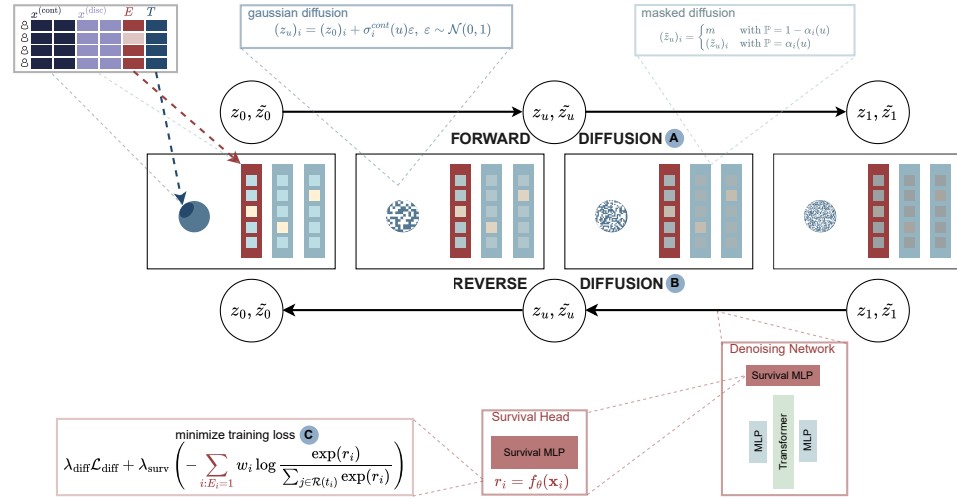

Figure 2: **Overview of our SURVDIFF**. SURVDIFF consisting of (A) *forward diffusion*, the (B) *backward diffusion* and the (C) *novel survival-focused loss*. Importantly, we distinguish the role of $E$ (event indicator; binary) and $T$ (time-to-event; continuous), which progress along different noising schemes due to the different variable types.

### 4.4 TRAINING AND SAMPLING

**Training:** SURVDIFF is trained end-to-end on minibatches. For each batch, we sample a noise level $u \sim U(0, 1)$ and corrupt the inputs via the forward processes. The network receives the noisy tuples, predicts denoised event indicators, event times, and continuous and discrete covariates, from which the diffusion loss $\mathcal{L}_{\text{diff}}$ is computed. Denoised covariates define the survival input, yield risk scores, and contribute to the survival loss $\mathcal{L}_{\text{surv}}$. To stabilize training $\lambda_{\text{surv}}$ is monotonically interpolated during a short warm-up period (Sønderby et al., 2016; Li et al., 2020) and then set to a calibrated value determined by *adaptive scaling*:

After a short calibration phase, the survival weight $\lambda_{\text{surv}}$ is chosen such that the survival term contributes a target fraction $\alpha_{\text{surv}}$ of the total objective, because the survival loss can differ substantially in scale across datasets. Using running averages $\bar{\mathcal{L}}_{\text{diff}}$ and $\bar{\mathcal{L}}_{\text{surv}}$ over the calibration window, the weight is computed as

$$\lambda_{\text{surv}} = \min\left\{\lambda_{\max}, \frac{\alpha_{\text{surv}}\bar{\mathcal{L}}_{\text{diff}}}{(1 - \alpha_{\text{surv}})(\bar{\mathcal{L}}_{\text{surv}} + \varepsilon)}\right\}. \tag{13}$$

This choice stabilizes the balance between diffusion and survival signals. The fixed calibrated weight preserves a stable training signal, as fully adaptive signals over all timesteps can drive the ratio by shrinking $\lambda_{\text{surv}}$ instead of minimizing the loss.

**Sampling:** After training we generate synthetic data $\mathcal{D}_{\text{syn}}$ by initializing continuous data points as $z_1 \sim \mathcal{N}(0, I)$ and discrete ones as $\tilde{z}_1 = m$, for $u = 1$. The learned reverse process then runs over a discretized schedule from $u = 1$ to $u = 0$, applying Gaussian denoising updates to $z_u$ and categorical unmasking to $\tilde{z}_u$. This yields a full synthetic sample $(x^{(\text{cont})}, x^{(\text{disc})}, E, T)$. Administrative censoring can be applied post hoc to reflect study-specific follow-up horizons.

## 5 EXPERIMENTS

We next evaluate SURVDIFF across multiple survival datasets and benchmarks, with all implementation details given in Supplement A. **Datasets:** We demonstrated the superior performance of SURVDIFF in extensive experiments across various medical datasets with *survival* data: (i) the ACTG clinical trial dataset (**AIDS**) (Hammer et al., 1997), (ii) the German Breast Cancer Study Group 2 dataset (**GBSG2**) (Schumacher et al., 1994), and (iii) the Molecular Taxonomy of Breast Cancer International Consortium dataset (**METABRIC**) (Pereira et al., 2016). Details for each dataset are in Supplement A.

| Metric | Method | AIDS | GBSG2 | METABRIC |
|--------|--------|------|-------|----------|
| **JS distance** (↓: better) | NFlow | 0.0129 ± 0.0017 | 0.0115 ± 0.0023 | 0.0123 ± 0.0017 |
| | TVAE | 0.0111 ± 0.0011 | 0.0130 ± 0.0009 | 0.0098 ± 0.0008 |
| | CTGAN | 0.0176 ± 0.0019 | 0.0120 ± 0.0017 | 0.0179 ± 0.0026 |
| | TabDiff | 0.0085 ± 0.0003 | 0.0179 ± 0.0005 | 0.0098 ± 0.0002 |
| | SurvivalGAN | 0.0135 ± 0.0018 | 0.0159 ± 0.0021 | 0.0212 ± 0.0027 |
| | Ashhad | 0.0074 ± 0.0003 | 0.0496 ± 0.0002 | 0.0070 ± 0.0010* |
| | SurvDiff (*ours*) | **0.0059 ± 0.0014** | **0.0074 ± 0.0007** | **0.0062 ± 0.0013** |
| **Wasserstein distance** (↓: better) | NFlow | 0.1161 ± 0.0106 | 0.0675 ± 0.0137 | 0.0826 ± 0.0144 |
| | TVAE | **0.0779 ± 0.0045** | 0.0400 ± 0.0033 | **0.0349 ± 0.0029** |
| | CTGAN | 0.2461 ± 0.0253 | 0.0558 ± 0.0094 | 0.1058 ± 0.0212 |
| | TabDiff | 0.0882 ± 0.0007 | 0.0533 ± 0.0012 | 0.0492 ± 0.0005 |
| | SurvivalGAN | 0.1545 ± 0.0151 | 0.0889 ± 0.0218 | 0.1689 ± 0.0272 |
| | Ashhad | 0.1068 ± 0.0021 | 0.9287 ± 0.0047 | 0.0890 ± 0.0040* |
| | SurvDiff (*ours*) | 0.0960 ± 0.0146 | **0.0347 ± 0.0026** | 0.0535 ± 0.0059 |
| **Shape error rate** (↓: better) | NFlow | 0.0858 ± 0.0104 | **0.1032 ± 0.0116** | 0.0872 ± 0.0106 |
| | TVAE | 0.0768 ± 0.0053 | 0.1403 ± 0.0051 | 0.0802 ± 0.0050 |
| | CTGAN | 0.1175 ± 0.0135 | 0.1260 ± 0.0140 | 0.1235 ± 0.0130 |
| | TabDiff | 0.0577 ± 0.0015 | 0.1392 ± 0.0038 | 0.0679 ± 0.0012 |
| | SurvivalGAN | 0.0934 ± 0.0083 | 0.1550 ± 0.0130 | 0.1507 ± 0.0168 |
| | Ashhad | 0.0983 ± 0.0034 | 0.2485 ± 0.0025 | * |
| | SurvDiff (*ours*) | **0.0494 ± 0.0134** | 0.1138 ± 0.0190 | **0.0519 ± 0.0121** |

*These values are the reported values in (Ashhad & Henao, 2024; 2025).

Table 2: **Covariate fidelity.** Covariate diversity metrics over different datasets (reported: mean ± s.d.) across 10 runs with different seeds).

**Baselines:** Our choice of benchmark is consistent with earlier work (Norcliffe et al., 2023). In particular, we benchmark our SURVDIFF against the following baselines for generating synthetic tabular or survival data: (1) **NFlow**, (2) **TVAE**, (3) **CTGAN**, (4) **TabDiff**, (5) **SurvivalGAN**, and (6) **Ashhad**. Details about the baselines and hyperparameters are in Supplement B.

**Performance metrics:** We compare the synthetic data along four dimensions:

(i) *Covariate fidelity.* We assess how closely the distribution of patient characteristics in the synthetic data matches the original data. For this, we compare the observed covariates via the Jensen-Shannon (JS) distance and the Wasserstein distance. We report marginal JS for per-feature alignment and joint WS to capture overall multivariate structure.

(ii) *Survival-specific fidelity.* We evaluate whether the synthetic data reproduce the temporal structure of the survival process. The evaluation includes the Event-Time Divergence (ETD) metric, which compares covariates of individuals with events occurring in similar equally sized time intervals (Supplement C), as well as temporal distribution plots for censored and uncensored events.

(iii) *Overall fidelity.* To assess fidelity across all patient variables, we report the Shape metric Shi et al. (2024b), which quantifies differences in the marginal distributions, and present normalized marginal histograms.

(iv) *Survival analysis performance.* The goal is to generate data that enable survival models trained on synthetic samples to generalize to real outcomes. For this, we train five popular survival models on the synthetic datasets, namely: (a) DeepHit (Lee et al., 2018), (b) Cox proportional hazards (Cox, 1972), (c) Weibull accelerated failure time regression (Weibull, 1951), (d) random survival forest (Ishwaran et al., 2008), and (e) XGBoost (Chen & Guestrin, 2016). We then compare the prediction quality on the real data with the corresponding model via: (1) the concordance index (C-index) (Harrell et al., 1982), which evaluates the accuracy of the ranking between predicted survival probabilities and observed event times, and (2) the Brier score (Brier, 1950), which assesses the calibration of the probabilistic predictions. We report averaged results across the five survival models over 10 different seeds.

# 6 RESULTS

• **Covariate fidelity:** We report the covariate diversity in Table 2. We observe the following: SURVDIFF consistently outperforms all other methods in terms of the marginal JS distance averaged over all features across all datasets. Furthermore, SURVDIFF achieves highly competitive performance measured by the joint Wasserstein distance in all experiments. SURVDIFF outperforms SurvivalGAN as the state-of-the-art baseline for synthetic survival data generation by a clear margin. For example, in

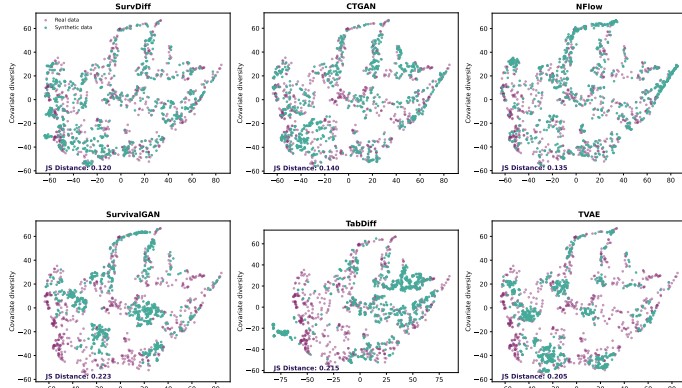

Figure 3: **t-SNE** visualization of covariate fidelity of real and synthetic data on GBSG2. ⇒ *Takeaway:* Synthetic samples from SURVDIFF are well aligned with the original data. SURVDIFF *achieves high covariate fidelity.*

terms of the joint WS, our SURVDIFF has a clearly lower distance compared to SurvivalGAN (GBSG2: −60%; etc.). Additional visualizations and implementation details are in Supplements A, D, and E.

*Insights:* To further evaluate the goodness-of-fit of the generated data, we visually assess the covariate fidelity in Fig. 3 and the survival-specific fidelity in Fig. 4. All baselines have large discrepancies between observed and synthetic covariates. This is particularly strong for SurvivalGAN, our main baseline, but also for other models. The results again confirm the fidelity of SURVDIFF.

• **Survival-specific fidelity:** We evaluate whether the synthetic data **preserve the temporal structure** of the survival process. Fig. 4 compares the event-time distributions for censored and uncensored patients. The curves show that patients who experience an event at early, mid, or late horizons exhibit **similar temporal patterns** in both the real and synthetic datasets. This indicates that SURVDIFF reproduces the progression of event times rather than collapsing toward frequent horizons.

• **Overall fidelity:** We further report the **Shape** metric (Shi et al., 2024b) in Tab. 2, which measures differences in the distributional shape of all patient variables and offers a focused view on whether real and synthetic samples share similar structural patterns. SURVDIFF achieves competitive performance. To evaluate specifically whether the time-to-event distribution is faithful, we explicitly report the Event-Time Divergence in Supplement C and normalized marginal covariate histograms in Supplement D, which further quantifies how well the synthetic data replicate characteristics of patients who experience an event at similar horizons.

*Insight.* In sum, SURVDIFF performs overall best in preserving the time-to-event dynamics and generating synthetic with high-fidelity temporal dynamics.

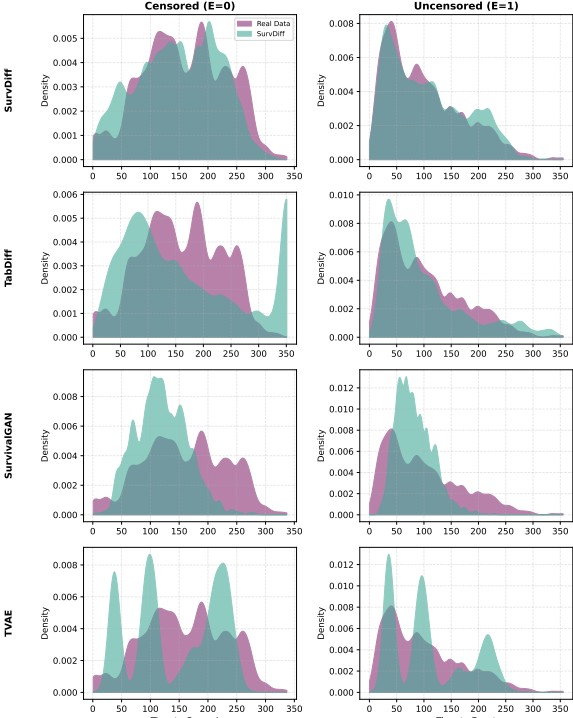

Figure 4: **Temporal distributions** of real and synthetic survival data on METABRIC, shown separately for censored and uncensored patients. ⇒ *Takeaway:* Synthetic patients from SURVDIFF exhibit similar event-time patterns as the real cohort, indicating strong temporal fidelity.

| Metric | Method | AIDS | GBSG2 | METABRIC |
|--------|--------|------|-------|----------|
| **C-Index** ($\uparrow$: better) | Real data | $0.6844 \pm 0.0925$ | $0.6592 \pm 0.0275$ | $0.6225 \pm 0.0225$ |
| | NFlow | $0.6032 \pm 0.0987$ | $0.6032 \pm 0.0987$ | $0.5711 \pm 0.0286$ |
| | TVAE | $0.6144 \pm 0.1018$ | $0.6406 \pm 0.0532$ | $0.5825 \pm 0.0531$ |
| | CTGAN | $0.5457 \pm 0.0205$ | $0.5945 \pm 0.0232$ | $0.5463 \pm 0.0310$ |
| | TabDiff | $0.6572 \pm 0.1117$ | $0.6286 \pm 0.0247$ | $\mathbf{0.6078 \pm 0.0144}$ |
| | SurvivalGAN | $0.6354 \pm 0.0553$ | $0.6357 \pm 0.0221$ | $0.5837 \pm 0.0092$ |
| | Ashhad | $0.5184 \pm 0.1324$ | $0.5062 \pm 0.0705$ | $0.5890 \pm 0.0150^{\dagger}$ |
| | SurvDiff (*ours*) | $\mathbf{0.7017 \pm 0.0782}$ | $\mathbf{0.6613 \pm 0.0215}$ | $0.5992 \pm 0.0276$ |
| **Brier Score** ($\downarrow$: better) | Real data | $0.0630 \pm 0.0013$ | $0.2063 \pm 0.0150$ | $0.1997 \pm 0.0114$ |
| | NFlow | $0.0532 \pm 0.0019$ | $0.2116 \pm 0.0083$ | $0.2109 \pm 0.0043$ |
| | CTGAN | $0.0671 \pm 0.0071$ | $0.2256 \pm 0.0025$ | $0.2477 \pm 0.0203$ |
| | TVAE | $0.0531 \pm 0.0015$ | $0.2115 \pm 0.0149$ | $0.2136 \pm 0.0082$ |
| | TabDiff | $0.0539 \pm 0.0052$ | $0.2130 \pm 0.0050$ | $\mathbf{0.1997 \pm 0.0086}$ |
| | SurvivalGAN | $0.0573 \pm 0.0026$ | $0.2154 \pm 0.0064$ | $0.2180 \pm 0.0055$ |
| | Ashhad | $0.0537 \pm 0.0021$ | $0.2192 \pm 0.0082$ | $0.2150 \pm 0.0050^{\dagger}$ |
| | SurvDiff (*ours*) | $\mathbf{0.0522 \pm 0.0024}$ | $\mathbf{0.2036 \pm 0.0092}$ | $0.2120 \pm 0.0040$ |

$^{\dagger}$ Values taken from Ashhad & Henao (2024; 2025).

Table 3: **Survival model performance.** Survival model metrics over different datasets (reported: mean $\pm$ s.d. across 10 runs with different seeds). $\Rightarrow$ *Takeaway:* Using synthetic samples from SURVDIFF consistently results in strong downstream performance results, especially *under strong right censoring*. Again, this benefit is especially *large* in comparison to the main baseline Survival-GAN.

• **Survival analysis performance:** In Table 3, we evaluate the performance of all models on downstream survival tasks. We observe that (1) SURVDIFF consistently achieves *large improvements* over SurvivalGAN and Ashhad on survival model tasks, (2) SURVDIFF achieves the best performance on AIDS and GBSG2, while performing on par with the best methods on METABRIC, and (3) the advantages of SURVDIFF are especially pronounced on datasets with stronger censoring (AIDS & GBSG2).

• **Sensitivity to dataset size:** Inspired by medical practice, we also present results on uniformly at random downsampled datasets to understand the sensitivity to small sample size settings, which are common in medicine. This additional sensitivity study is presented in Supplement G). Therein, we see *large benefits of* SURVDIFF *over existing methods in small-sample settings*. Hence, our method is well-designed to meet needs in medical practice.

• **Additional results:** For completeness, we also report Kaplan-Meier-based metrics in Supplement F. Therein, SURVDIFF shows comparable performance. We further include ablation studies and parameter sensitivity analysis of our novel loss in Supplement J, and visualize loss convergence in Supplement H.

• **Extension to differential privacy:** We show that SURVDIFF can be readily extended to incorporate differential privacy. For this, we present a differentially private variant of SURVDIFF, which offers formal privacy guarantees under DP-SGD (Dwork & Roth, 2014; Abadi et al., 2016). Implementation details and experiment results are in Supplement I. We show that SURVDIFF outperforms the DP-GAN baseline across covariate fidelity and survival analysis performance metrics.

# 7 DISCUSSION

**Clinical considerations.** We follow needs in clinical research, where it is essential to preserve patient characteristics in synthetic data (Yan et al., 2022; Giuffrè & Shung, 2023). Existing baselines, such as SurvivalGAN, often fail to do so, leading to mismatches that no longer accurately reflect the true patient population. Since summarizing patient demographics is typically the first step in clinical studies, inaccuracies in the patient covariate distributions are particularly problematic: they can distort estimates of incidence rates and lead to misleading subgroup survival times. Hence, a key strength of our method is to preserve covariate fidelity; i.e., ensuring that synthetic datasets remain clinically meaningful while also supporting strong survival analysis performance.

**Conclusion:** We propose SURVDIFF, a novel end-to-end diffusion model tailored to generating survival data. Our SURVDIFF jointly generates patient covariates, event times, and right-censoring indicators in an end-to-end manner. As a result, SURVDIFF generating reliable synthetic datasets that (i) match patient characteristics and (ii) produce faithful event-time distributions that preserve censoring mechanisms and thus improve downstream survival analysis.

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

# A    IMPLEMENTATION DETAILS

## A.1    DATASETS

**AIDS (ACTG 320 Trial).** The AIDS dataset[2] originates from the ACTG 320 trial, which evaluated combination antiretroviral therapy in HIV patients (Hammer et al., 1997). It contains data from 1151 patients. The observed event is death, and $91.7\%$ of patients are censored. Covariates include baseline clinical and laboratory measures such as CD4 cell count, age, hemoglobin, weight, and prior therapy indicators.

**GBSG2 (German Breast Cancer Study Group 2).** The GBSG2 dataset[3] stems from a randomized clinical trial of 686 breast cancer patients treated between 1984 and 1989 (Schumacher et al., 1994). The endpoint is recurrence-free survival, defined as the time to relapse or death, whichever occurs first. Here, $56.4\%$ patients are censored. Covariates cover age, menopausal status, tumor size, grade, number of positive lymph nodes, progesterone and estrogen receptor levels, and hormone therapy status.

**METABRIC (Molecular Taxonomy of Breast Cancer International Consortium).** The METABRIC dataset[4] is a large breast cancer cohort study with 1903 patients and long-term follow-up (Pereira et al., 2016). The event of interest is overall survival. The censoring rate is $42\%$. It includes a mix of clinical variables (age, tumor size, grade, receptor status).

## A.2    IMPLEMENTATION OF SURVDIFF

SURVDIFF is implemented in Pytorch. All experiments were carried out on one NVIDIA A100-PCIE-40GB. The default settings of our method and all benchmarking methods are listed below in Section B. The model architecture is based on the architecture of (Shi et al., 2024b). Each of the experiments was concluded after at most 13min.

Covariates: We embed high-cardinality discrete covariates as continuous vectors; however, we still distinguish them formally by their underlying finite support.

---

[2]`https://scikit-survival.readthedocs.io/en/stable/api/generated/sksurv.datasets.load_aids.html`
[3]`https://scikit-survival.readthedocs.io/en/stable/api/generated/sksurv.datasets.load_gbsg2.html`
[4]`https://github.com/havakv/pycox`

# B  HYPERPARAMETERS

The hyperparameter grids for NFlow, CTGAN, TVAE, and SurvivalGAN follow the configurations in the SurvivalGAN paper (Norcliffe et al., 2023) provided in the SynthCity library (Qian et al., 2023). For the Ashhad baseline, we use the hyperparameters reported in the original paper (Ashhad & Henao, 2024; 2025). All benchmark models are run with these published settings to ensure comparability across datasets.

| Model | Hyperparameters | |
|---|---|---|
| | No. Epochs | 1200 |
| | Transformer Hidden Layers | 5 |
| | MLP Hidden Layers | 3 |
| | Survival MLP Hidden Layers | 2 |
| | $\sigma_{\min}$ | 0.002 |
| | $\sigma_{\max}$ | 20.0 |
| SURVDIFF | Learning Rate | 0.001 |
| | Weight Decay | 0.0001 |
| | Dropout | 0.1 |
| | Batch Size | 256 |
| | Warm-up Epochs | 150 |
| | $\alpha_{\mathrm{surv}}$ | 0.3 |
| | Calibration Steps | 10 |
| | Sampling Steps | 300 |

Table 4: Hyperparameters for SURVDIFF.

| Model | Hyperparameters | |
|---|---|---|
| CoxPH | Estimation Method | Breslow |
| | Penalizer | 0.0 |
| | $L^1$ Ratio | 0.0 |
| Weibull AFT | $\alpha$ | 0.05 |
| | Penalizer | 0.0 |
| | $L^1$ Ratio | 0.0 |
| SurvivalXGBoost | Objective | Survival: AFT |
| | Evaluation Metric | AFT Negative Log Likelihood |
| | AFT Loss Distribution | Normal |
| | AFT Loss Distribution Scale | 1.0 |
| | No. Estimators | 100 |
| | Column Subsample Ratio (by node) | 0.5 |
| | Maximum Depth | 8 |
| | Subsample Ratio | 0.5 |
| | Learning Rate | 0.05 |
| | Minimum Child Weight | 50 |
| | Tree Method | Histogram |
| | Booster | Dart |
| RandomSurvivalForest | Max Depth | 3 |
| | No. Estimators | 100 |
| | Criterion | Gini |
| Deephit | No. Durations | 1000 |
| | Batch Size | 100 |
| | Epochs | 2000 |
| | Learning Rate | 0.001 |
| | Hidden Width | 300 |
| | $\alpha$ | 0.28 |
| | $\sigma$ | 0.38 |
| | Dropout Rate | 0.02 |
| | Patience | 20 |
| | Using Batch Normalization | True |

Table 5: Hyperparameters for survival models.

| Model | Hyperparameters | |
|---|---|---|
| | CTGAN | |
| | No. Iterations | 1500 |
| | Generator Hidden Layers | 3 |
| | Discriminator Hidden Layers | 2 |
| | Discriminator and Generator Hidden Width | 250 |
| | Discriminator Non-linearity | Leaky ReLU |
| | Generator Non-linearity | Tanh |
| | Discriminator and Generator Dropout Rate | 0.1 |
| | Learning Rate | 0.001 |
| | Weight Decay | 0.001 |
| | Batch Size | 500 |
| | Gradient Penalty ($\lambda$ | 10 |
| | Encoder Max Clusters | 10 |
| | DeepHit | |
| SurvivalGAN | No. Durations | 100 |
| | Batch Size | 100 |
| | No. Epochs | 2000 |
| | Learning Rate | 0.001 |
| | Hidden Width | 300 |
| | $\alpha$ | 0.28 |
| | $\sigma$ | 0.38 |
| | Dropout Rate | 0.02 |
| | Patience | 20 |
| | Using Batch Normalization | True |
| | XGBoost | |
| | No. Estimators | 200 |
| | Depth | 5 |
| | Booster | Dart |
| | Tree Method | Histogram |
| | No. Epochs | 4000 |
| | Transformer Hidden Layers | 5 |
| | MLP Hidden Layers | 2 |
| TabDiff | $\sigma_{min}$ | 0.002 |
| | $\sigma_{max}$ | 80.0 |
| | Learning Rate | 0.002 |
| | Batch Size | 256 |
| | Sampling Steps | 300 |
| | Embedding Width | 10 |
| | Generator and Discriminator No. Hidden Layers | 2 |
| | Generator and Discriminator Hidden Width | 256 |
| | Generator and Discriminator Learning Rate | $2 \times 10^{-4}$ |
| CTGAN | Generator and Discriminator Decay | $1 \times 10^{-6}$ |
| | Batch Size | 500 |
| | Discriminator Steps | 1 |
| | No. Iterations | 300 |
| | Pac | 10 |
| | Embedding Width | 128 |
| | Encoder and Decoder No. Hidden Layers | 2 |
| | Encoder and Decoder Hidden Width | 128 |
| TVAE | $L^2$ Scale | $1 \times 10^{-5}$ |
| | Batch Size | 500 |
| | No. Iterations | 300 |
| | Loss Factor | 2 |
| | No. Iterations | 500 |
| | No. Hidden Layers | 1 |
| | Hidden Width | 100 |
| | Batch Size | 100 |
| | No. Transform Blocks | 1 |
| | Dropout Rate | 0.1 |
| NFlow | No. Bins | 8 |
| | Tail Bound | 3 |
| | Learning Rate | $1 \times 10^{-3}$ |
| | Base Distribution | Standard Normal |
| | Linear Transform Type | Permutation |
| | Base Transform Type | Affine-Coupling |
| | No. Iterations | 1000 |
| | Batch Size | 1024 |
| | Learning Rate | 0.002 |
| Ashhad | Weight Decay | 0.0001 |
| | No. of Time-Steps | 1000 |
| | Scheduler | Cosine |
| | Gaussian Loss Type | MSE |

Table 6: Hyperparameters for benchmarking models.

### B.1 IMPLEMENTATION DETAILS OF THE COVARIATE DISTRIBUTION EXPERIMENTS

For each dataset with $n$ training samples and $p$ covariates, we generate exactly $n$ synthetic samples with every method. All covariates are preprocessed *exactly in the same way* as for the survival models (continuous features standardized, categorical variables one-hot encoded), and we do *not* apply any additional dimensionality reduction (no PCA or similar). Following the commonly used SynthCity library (Qian et al., 2023), we report a marginal Jensen–Shannon distance and a joint Wasserstein distance.

**Jensen–Shannon distance (marginal).** For each covariate $k \in \{1, \ldots, p\}$ we approximate its marginal distribution on the real data by an equal–width histogram with

$$B = \min\{10, \ \#\{\text{unique values in } X_{\text{real}}^{(k)}\}\} \tag{14}$$

bins. The resulting bin edges are reused to bin the synthetic data, so real and synthetic histograms share the same support. Let $p^{(k)}$ and $q^{(k)}$ denote the corresponding normalized bin counts. We apply add–one smoothing to all bins and compute the Jensen–Shannon distance $\text{JSD}(p^{(k)}, q^{(k)})$ using the SciPy implementation. The reported value is the average over covariates, i.e.,

$$\text{JSD}_{\text{marginal}} = \frac{1}{p} \sum_{k=1}^{p} \text{JSD}(p^{(k)}, q^{(k)}). \tag{15}$$

After preprocessing, there are no missing values; extreme observations are not removed but simply fall into the outermost histogram bins.

**Wasserstein distance (joint).** Let $X \in \mathbb{R}^{n \times p}$ and $\tilde{X} \in \mathbb{R}^{n \times p}$ denote the real and synthetic covariate matrices, respectively. We apply feature-wise min-max scaling to $[0, 1]$ using only the real data via

$$\hat{X} = \text{MinMax}(X), \qquad \hat{\tilde{X}} = \text{MinMax}(\tilde{X}), \tag{16}$$

and treat $\hat{X}$ and $\hat{\tilde{X}}$ as empirical distributions over $\mathbb{R}^p$ with equal mass $1/n$ on each sample. We then compute a Sinkhorn–regularized 2–Wasserstein distance using the `SamplesLoss(loss="sinkhorn")` optimal transport solver (GeomLoss). This matches the `WassersteinDistance` metric in SynthCity (Qian et al., 2023). Since all datasets are fully observed after preprocessing, NaNs do not occur, and potential anomalies are handled solely through the min–max scaling.

## C  EVENT-TIME DIVERGENCE (ETD)

A survival-aware generative model should reproduce not only when events occur but also which types of patients tend to experience events at different stages of the disease course. If the synthetic cohort is to be useful for survival modeling, then the synthetic patients who die early, mid-course, or late should resemble the corresponding groups in the real cohort. The Event-Time Divergence (ETD) metric evaluates this alignment.

We divide the observed event-time horizon into five equally sized intervals and focus on uncensored individuals whose events fall within each interval. For every interval, we compare the covariate distribution of real patients who die in that interval with that of synthetic patients whose generated event times fall in the same interval. The comparison uses the Jensen-Shannon distance, producing five divergence scores that measure how well the model reproduced the covariate composition of event-time matched subpopulations.

Formally, we have

$$ETD = \sum_{k=1}^{5} JSDist(P_{\text{real}}(X \mid E = 1, T \in \mathcal{I}_k), P_{\text{syn}}(X \mid E = 1, T \in \mathcal{I}_k), \tag{17}$$

where $\mathcal{I}_k$ denotes the $k$-th of five equal-mass event-time intervals obtained by partitioning uncensored event times $T$. We then aggregate these per-interval divergences into a sum across intervals.

Across all datasets, SURVDIFF yields the lowest ETD values for the aggregated metric (Tables 7–9). This reflects that our model not only captures the overall covariates structure but also generates patients with event-time patterns that mirror those of real clinical cohorts.

| Event-Time-Divergence | Ctgan | Tvae | Nflow | Survival_gan | Tabdiff | Survdiff |
|---|---|---|---|---|---|---|
| $0 \leq T \leq 60.4$ | $0.0672 \pm 0.0158$ | $0.0515 \pm 0.0193$ | $0.0348 \pm 0.0078$ | $0.0360 \pm 0.0133$ | $0.0245 \pm 0.0034$ | $0.0253 \pm 0.0049$ |
| $60.4 < T \leq 119.8$ | $0.0610 \pm 0.0103$ | $0.0264 \pm 0.0044$ | $0.0349 \pm 0.0093$ | $0.0331 \pm 0.0158$ | $0.0313 \pm 0.0029$ | $0.0291 \pm 0.0031$ |
| $119.8 < T \leq 179.2$ | $0.0640 \pm 0.0149$ | $0.0697 \pm 0.0078$ | $0.0347 \pm 0.0093$ | $0.0414 \pm 0.0103$ | $0.0377 \pm 0.0041$ | $0.0404 \pm 0.0062$ |
| $179.2 < T \leq 238.64$ | $0.0647 \pm 0.0182$ | $0.0479 \pm 0.0073$ | $0.0409 \pm 0.0077$ | $0.0583 \pm 0.0111$ | $0.0523 \pm 0.0044$ | $0.0458 \pm 0.0032$ |
| $238.64 < T \leq 298.0$ | $0.0613 \pm 0.0107$ | $0.0304 \pm 0.0043$ | $0.0306 \pm 0.0102$ | $0.0854 \pm 0.0000$ | $0.0616 \pm 0.0107$ | $0.0284 \pm 0.0046$ |
| sum | $0.3182 \pm 0.0320$ | $0.2296 \pm 0.0229$ | $0.1813 \pm 0.0199$ | $0.2532 \pm 0.0256$ | $0.2073 \pm 0.0192$ | $0.1690 \pm 0.0102$ |

Table 7: **Event-Time-Divergence on AIDS.** For five equally sized event-time intervals, we compute the Jensen–Shannon distance between real and synthetic distributions *using only uncensored individuals who die within each interval*, ensuring covariate-matched comparison. Reported: mean $\pm$ s.d. across runs.

| Event-Time-Divergence | Ctgan | Tvae | Nflow | Survival_gan | Tabdiff | Survdiff |
|---|---|---|---|---|---|---|
| $0 \leq T \leq 548.8$ | $0.0266 \pm 0.0091$ | $0.0182 \pm 0.0040$ | $0.0196 \pm 0.0035$ | $0.0270 \pm 0.0077$ | $0.0329 \pm 0.0021$ | $0.0192 \pm 0.0022$ |
| $548.8 < T \leq 1025.6$ | $0.0230 \pm 0.0049$ | $0.0157 \pm 0.0019$ | $0.0166 \pm 0.0038$ | $0.0262 \pm 0.0073$ | $0.0231 \pm 0.0001$ | $0.0134 \pm 0.0035$ |
| $1025.6 < T \leq 1502.4$ | $0.0253 \pm 0.0047$ | $0.0267 \pm 0.0031$ | $0.0222 \pm 0.0050$ | $0.0350 \pm 0.0084$ | $0.0295 \pm 0.0022$ | $0.0194 \pm 0.0026$ |
| $1502.4 < T \leq 1979.2$ | $0.0299 \pm 0.0053$ | $0.0250 \pm 0.0032$ | $0.0272 \pm 0.0060$ | $0.0502 \pm 0.0116$ | $0.0349 \pm 0.0048$ | $0.0239 \pm 0.0017$ |
| $1979.2 < T \leq 2456.0$ | $0.0506 \pm 0.0125$ | $0.0607 \pm 0.0161$ | $0.0384 \pm 0.0048$ | $0.0870 \pm 0.0066$ | $0.0436 \pm 0.0071$ | $0.0341 \pm 0.0069$ |
| sum | $0.1553 \pm 0.0177$ | $0.1463 \pm 0.0173$ | $0.1214 \pm 0.0105$ | $0.2255 \pm 0.0190$ | $0.1641 \pm 0.0093$ | $0.1100 \pm 0.0086$ |

Table 8: **Event-Time-Divergence on GBSG2.** For five equally sized event-time intervals, we compute the Jensen–Shannon distance between real and synthetic distributions *using only uncensored individuals who die within each interval*, ensuring covariate-matched comparison. Reported: mean $\pm$ s.d. across runs.

| Event-Time-Divergence | Ctgan | Tvae | Nflow | Survival_gan | Tabdiff | Survdiff |
|---|---|---|---|---|---|---|
| $0 \leq T \leq 71.1$ | $0.0280 \pm 0.0081$ | $0.0162 \pm 0.0029$ | $0.0173 \pm 0.0042$ | $0.0310 \pm 0.0108$ | $0.0158 \pm 0.0007$ | $0.0130 \pm 0.0030$ |
| $71.1 < T \leq 142.1$ | $0.0292 \pm 0.0119$ | $0.0134 \pm 0.0022$ | $0.0129 \pm 0.0036$ | $0.0287 \pm 0.0038$ | $0.0131 \pm 0.0009$ | $0.0089 \pm 0.0010$ |
| $142.1 < T \leq 213.2$ | $0.0276 \pm 0.0082$ | $0.0128 \pm 0.0016$ | $0.0162 \pm 0.0045$ | $0.0328 \pm 0.0094$ | $0.0196 \pm 0.0021$ | $0.0108 \pm 0.0013$ |
| $213.2 < T \leq 284.2$ | $0.0288 \pm 0.0078$ | $0.0193 \pm 0.0033$ | $0.0202 \pm 0.0029$ | $0.0645 \pm 0.0108$ | $0.0221 \pm 0.0015$ | $0.0165 \pm 0.0034$ |
| $284.2 < T \leq 355.2$ | $0.0600 \pm 0.0189$ | $0.0732 \pm 0.0009$ | $0.0425 \pm 0.0084$ | $0.0722 \pm 0.0000$ | $0.0279 \pm 0.0029$ | $0.0363 \pm 0.0112$ |
| sum | $0.1735 \pm 0.0263$ | $0.1349 \pm 0.0053$ | $0.1092 \pm 0.0114$ | $0.2291 \pm 0.0183$ | $0.0985 \pm 0.0041$ | $0.0855 \pm 0.0122$ |

Table 9: **Event-Time-Divergence on METABRIC.** For five equally sized event-time intervals, we compute the Jensen–Shannon distance between real and synthetic distributions *using only uncensored individuals who die within each interval*, ensuring covariate-matched comparison. Reported: mean $\pm$ s.d. across runs.

# D    ADDITIONAL COVARIATE, EVENT-TIME, EVENT-INDICATOR, AND KAPLAN-MEIER VISUALIZATIONS

To complement the main results, we provide additional visualizations of covariate, event-time, and event-indicator structure across all datasets. Figures 5 and 3 report t-SNE embeddings comparing real and synthetic covariates for the baseline models on the AIDS and METABRIC datasets. Figures 10–12 present joint t-SNE and Kaplan-Meier visualizations for SURVDIFF and baselines, aggregated over ten random seeds, illustrating alignment in covariate geometry and Kaplan-Meier trajectories. Finally, Figures 7–9 show marginal distributions for all covariates, offering a complementary view of univariate fidelity. Together, these visualizations provide a qualitative assessment of the stability of training and the consistency of generated covariates and event-time characteristics across datasets.

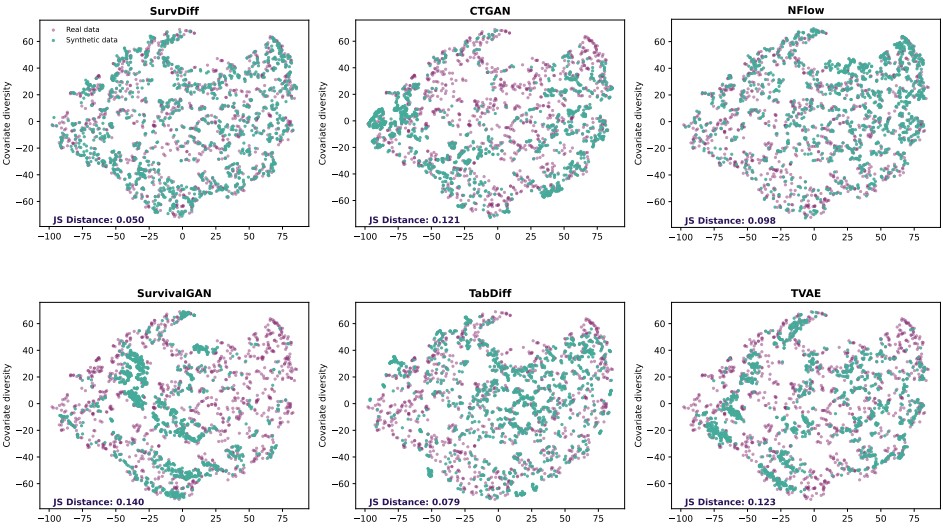

Figure 5: **t-SNE** visualization of covariate fidelity on the AIDS dataset.

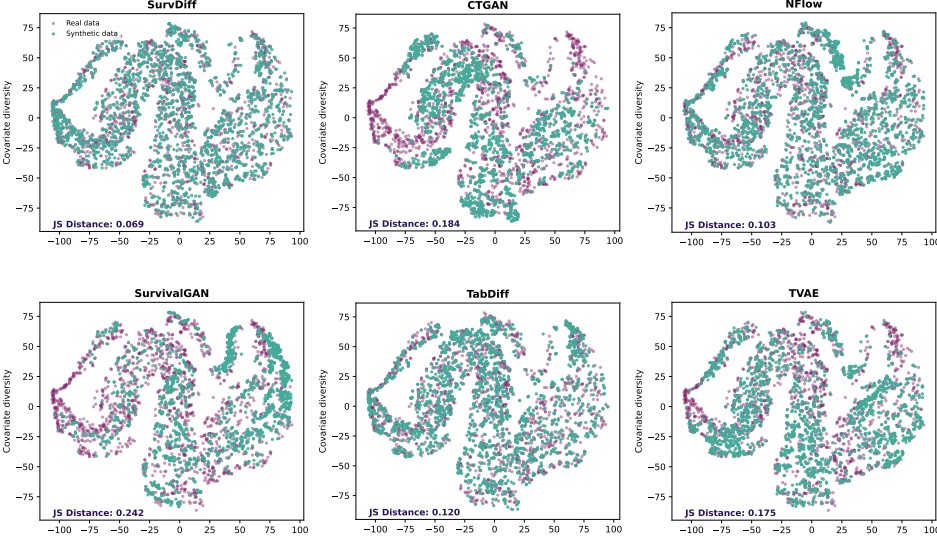

Figure 6: **t-SNE** visualization of covariate fidelity on the METABRIC dataset.

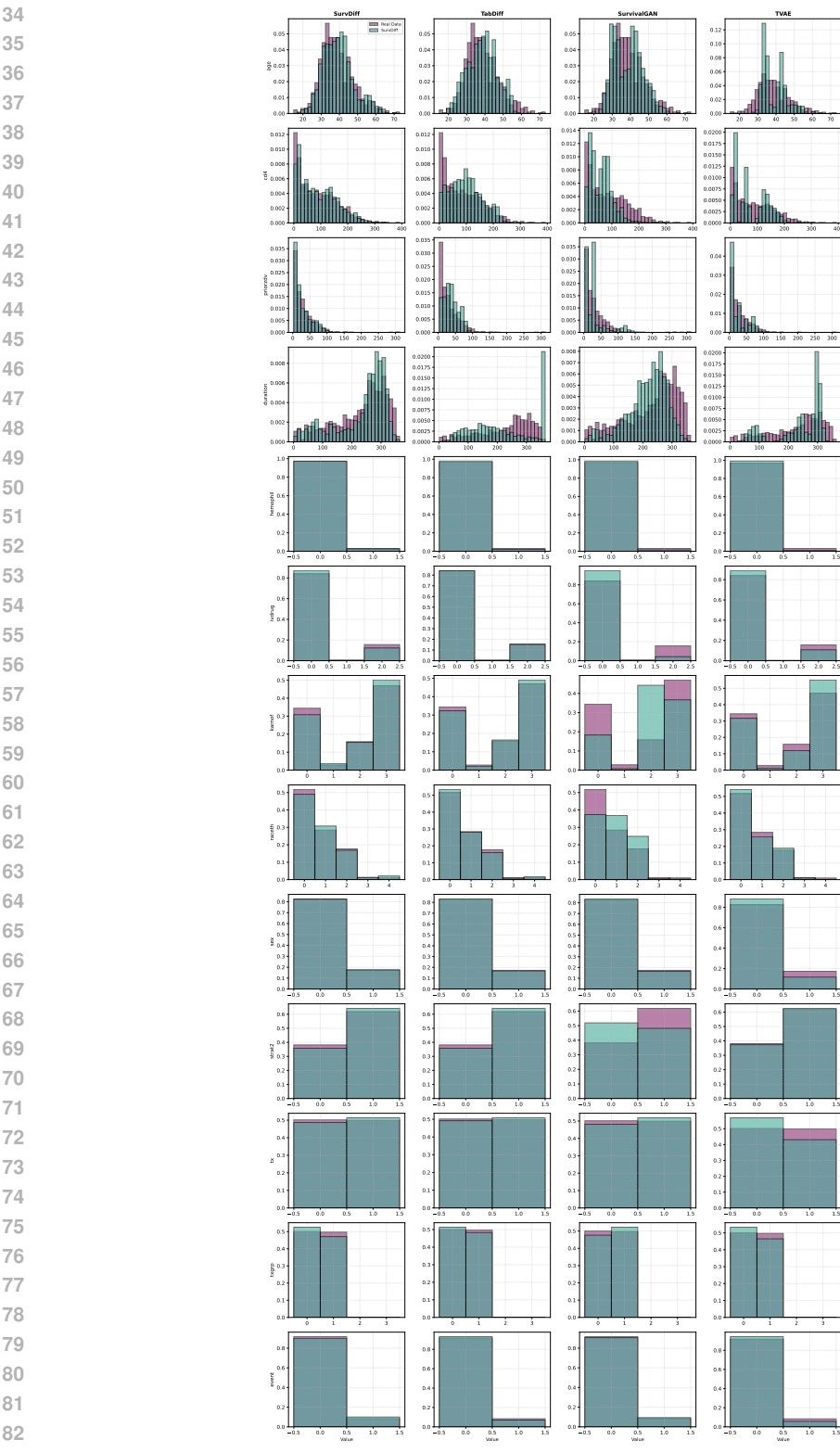

Figure 7: **Marginal probability** visualization on the AIDS dataset.

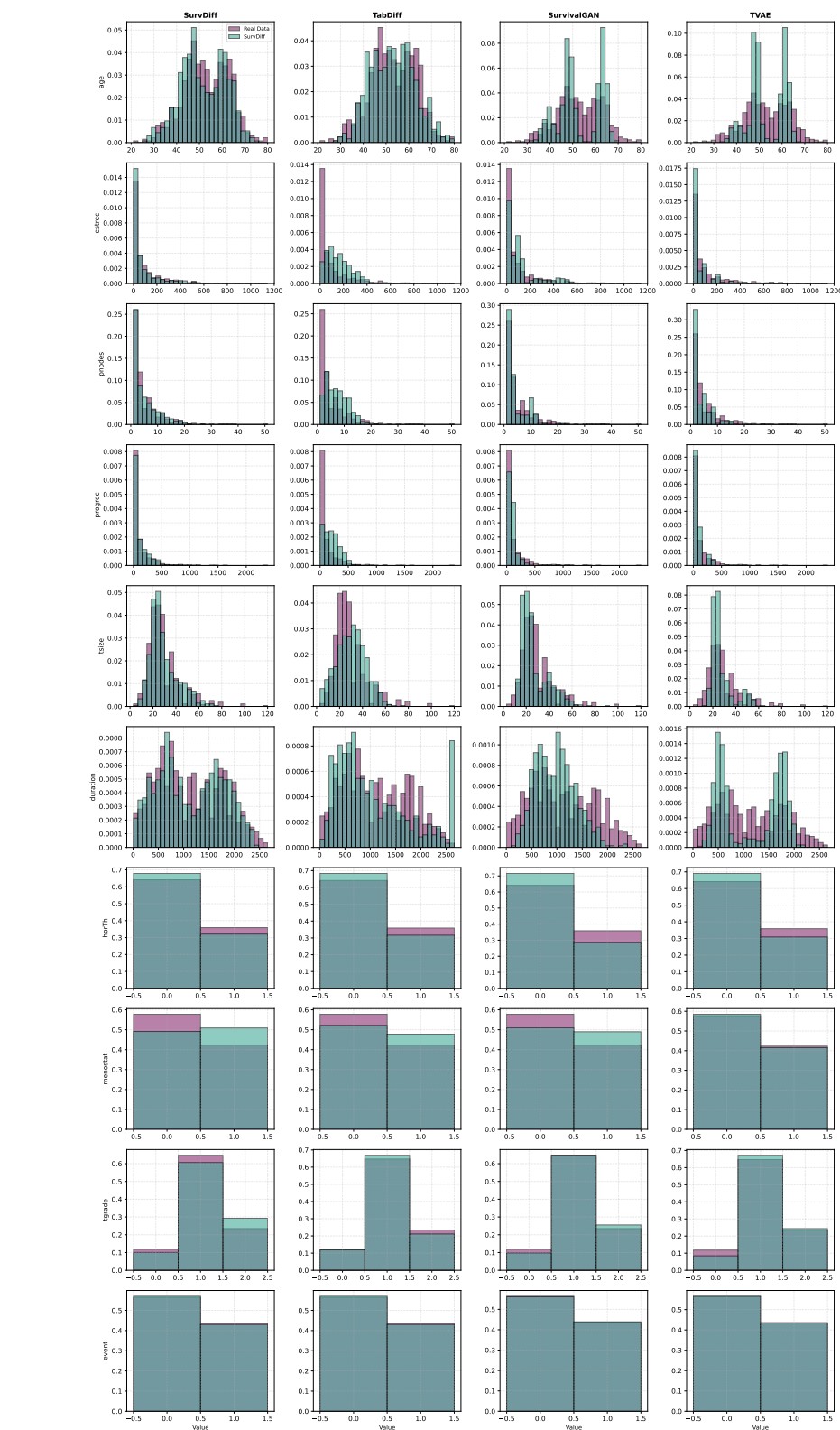

Figure 8: **Marginal probability** visualization on the GBSG2 dataset.

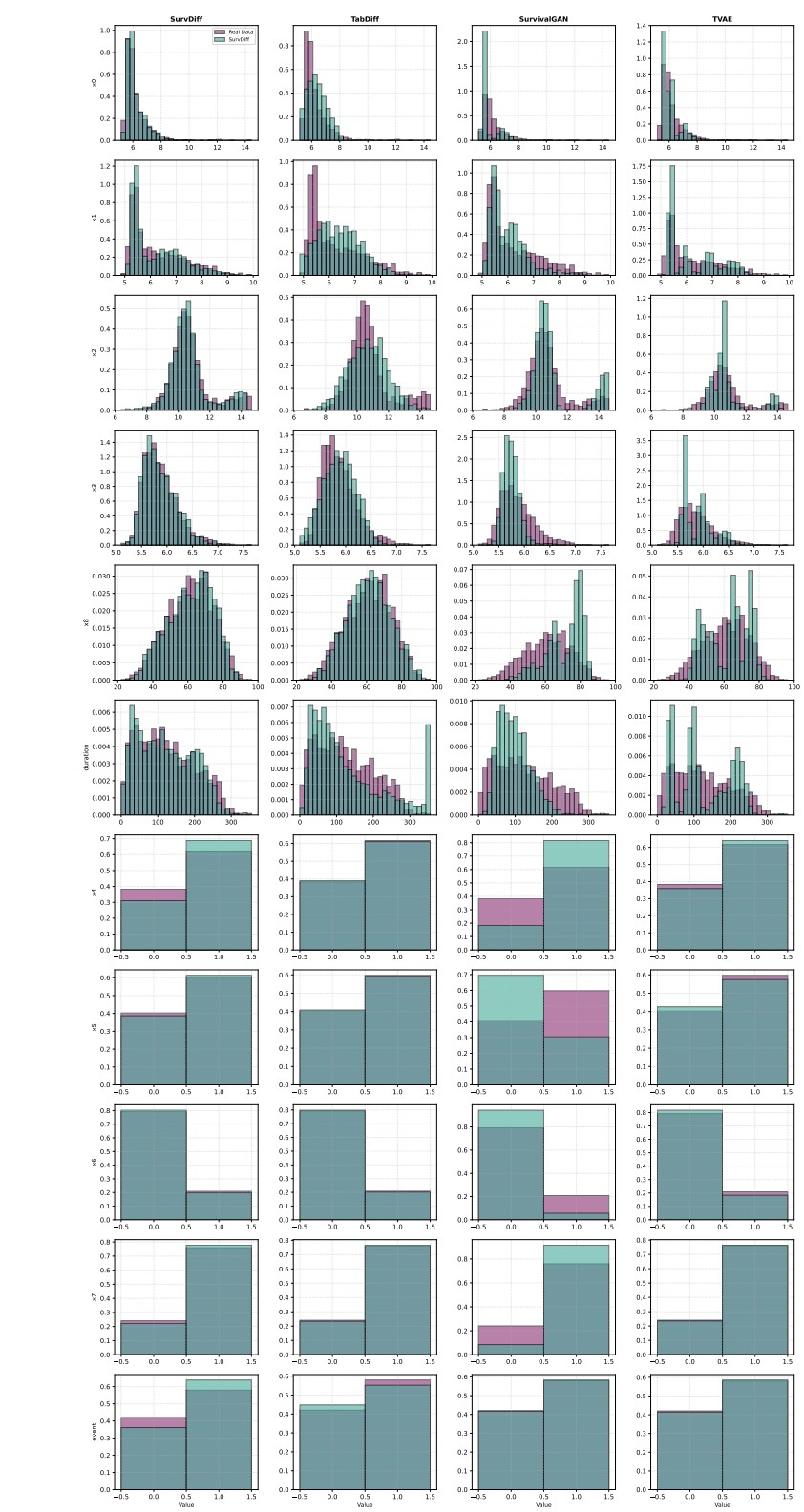

Figure 9: **Marginal probability** visualization on the METABRIC dataset.

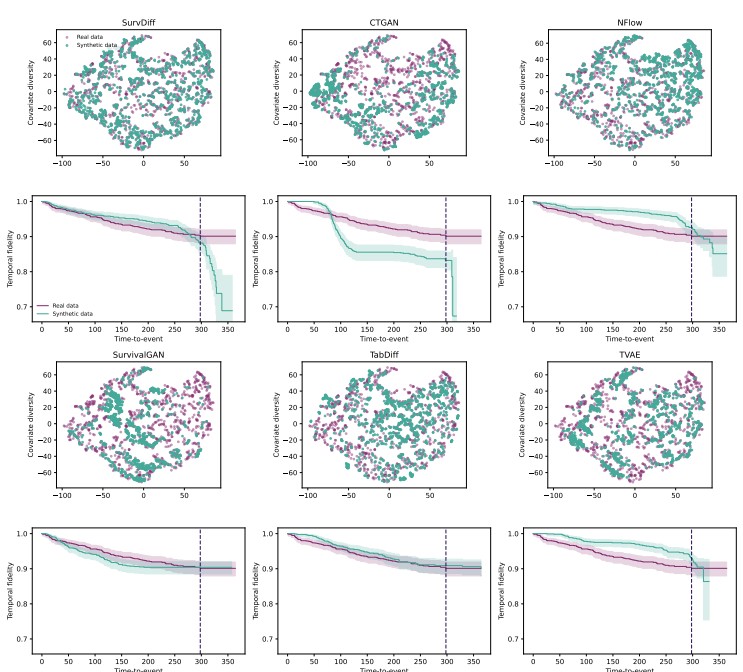

Figure 10: **t-SNE** visualization and **KM** curves on the AIDS dataset.

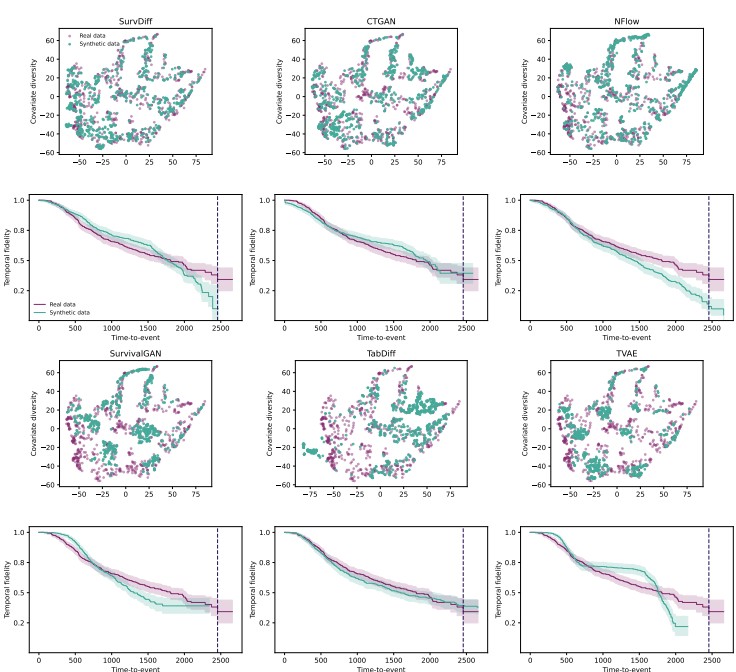

Figure 11: **t-SNE** visualization and **KM** curves on the GBSG2 dataset.

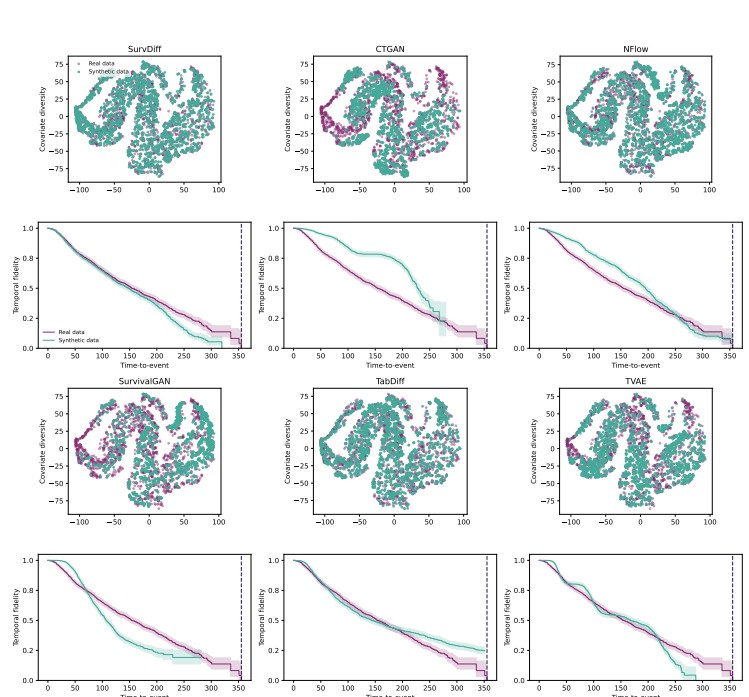

Figure 12: **t-SNE** visualization and **KM** curves on the METABRIC dataset.

## E  ADDITIONAL TEMPORAL SURVIVAL-TIME VISUALIZATIONS

To assess how well the generative models reproduce temporal survival structure, we report time-to-censoring and time-to-event distributions in Figures 13 and 14. These density plots compare the empirical survival times of real individuals with those generated by each baseline model, separately for censored and uncensored cases. The visualizations highlight whether synthetic cohorts capture early-event behavior, late-event tails, and typical censoring patterns observed in the real data. Together, these plots provide a qualitative view of temporal fidelity that complements the Event-Time Divergence (ETD) metric in Supplement C and the results reported in the main text.

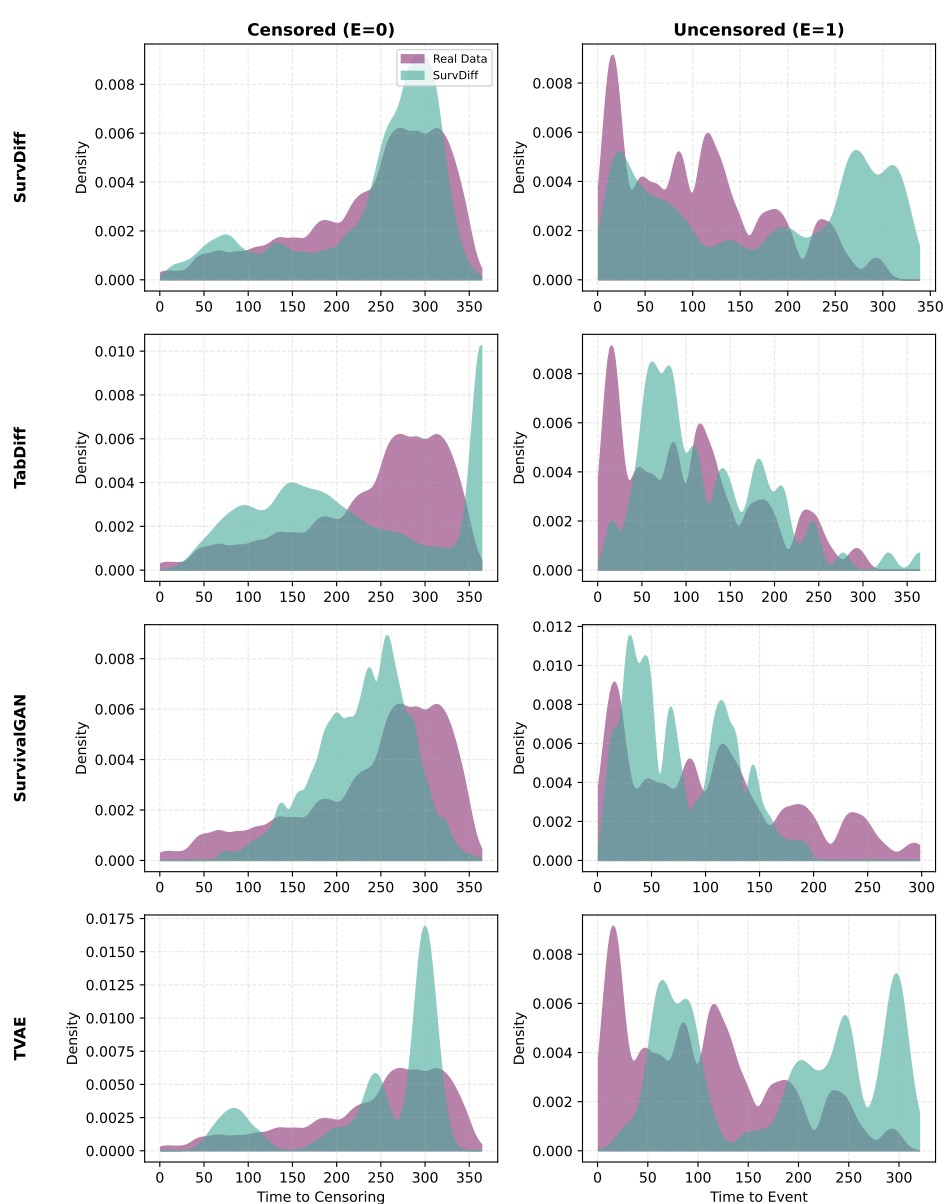

Figure 13: **Temporal fidelity** visualization of covariate fidelity on the AIDS dataset.

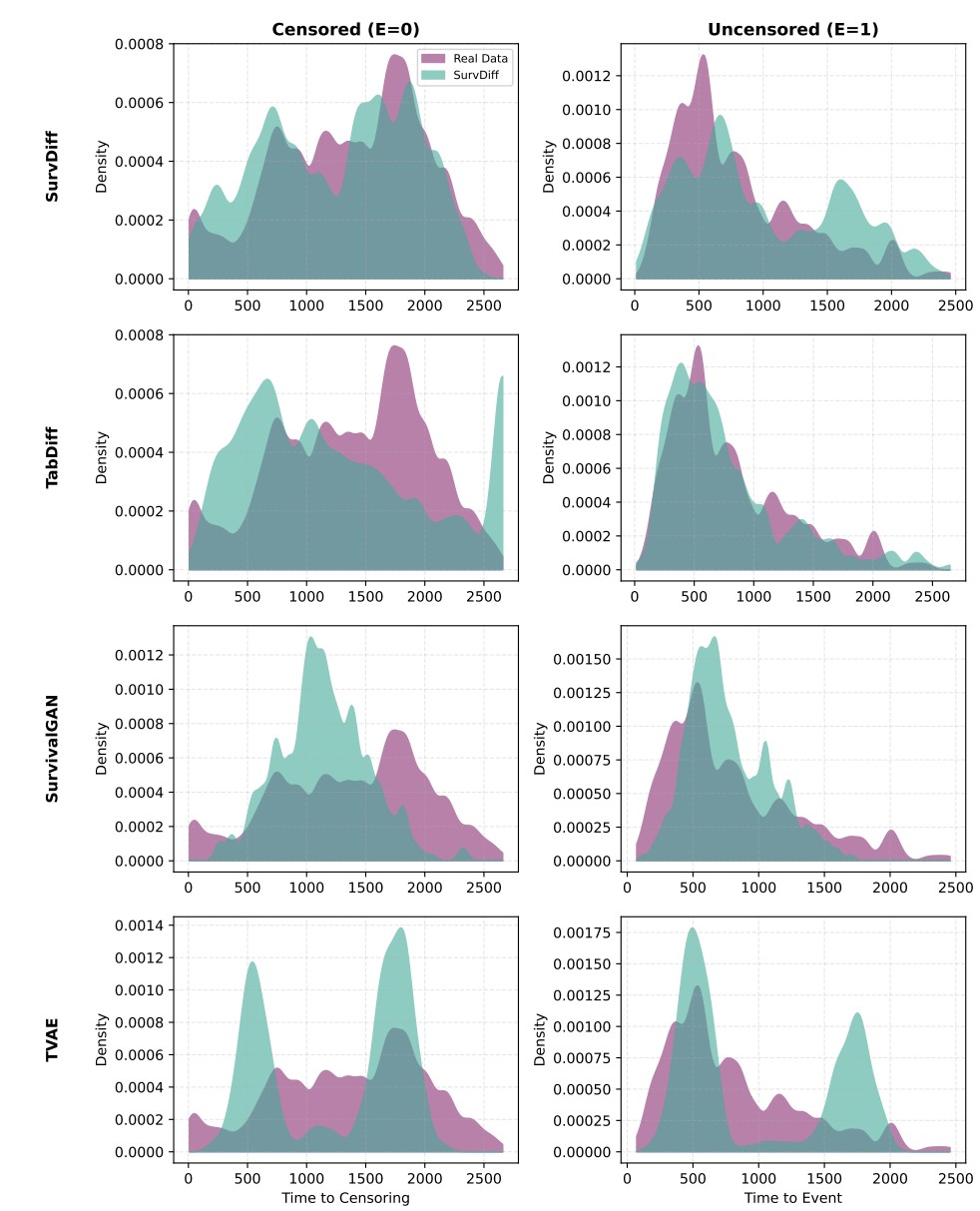

Figure 14: **Temporal fidelity** visualization of covariate fidelity on the GBSG2 dataset.

# F KAPLAN-MEIER METRICS

In addition to the (i) *covariate distribution fidelity* metrics and the (ii) *survival model performance* metrics, we also examine (iii) *survival* metrics, where SURVDIFF shows broadly comparable performance across datasets. We evaluate how well synthetic data reproduce survival outcomes. For this, we compare Kaplan-Meier curves (Kaplan & Meier, 1958) of real and synthetic cohorts using the mean squared error (KM MSE) (Fay et al., 2013), and quantify differences in restricted mean survival time (RMST gap) (Royston & Parmar, 2011; Kim et al., 2017) up to a fixed horizon. For the RMST gap, it is important to note that, since it summarizes the difference in areas under the survival curves, it can mask deviations that cancel each other out (e.g., synthetic survival curves slightly above real ones early but below later). This is shown in Table 10. Overall, the results show the strong performance of our SURVDIFF.

| Metric | Method | AIDS | GBSG2 | METABRIC |
|---|---|---|---|---|
| **RMST gap** (↓: better) | NFlow | $0.0235 \pm 0.0066$ | $0.0697 \pm 0.0205$ | $0.0598 \pm 0.0251$ |
| | TVAE | $0.0360 \pm 0.0023$ | $0.0408 \pm 0.0152$ | $\mathbf{0.0223 \pm 0.0064}$ |
| | CTGAN | $0.0491 \pm 0.0204$ | $0.0510 \pm 0.0178$ | $0.0890 \pm 0.0248$ |
| | TabDiff | $0.0092 \pm 0.0014$ | $\mathbf{0.0091 \pm 0.0034}$ | $0.0329 \pm 0.0027$ |
| | SurvivalGAN | $\mathbf{0.0079 \pm 0.0028}$ | $0.0319 \pm 0.0124$ | $0.0644 \pm 0.0178$ |
| | SurvDiff (*ours*) | $0.0155 \pm 0.0051$ | $0.0412 \pm 0.0208$ | $0.0577 \pm 0.0267$ |
| **KM MSE** (↓: better) | NFlow | $0.0009 \pm 0.0003$ | $0.0095 \pm 0.0042$ | $0.0082 \pm 0.0036$ |
| | TVAE | $0.0015 \pm 0.0002$ | $0.0109 \pm 0.0027$ | $\mathbf{0.0036 \pm 0.0006}$ |
| | CTGAN | $0.0049 \pm 0.0041$ | $0.0087 \pm 0.0077$ | $0.0109 \pm 0.0027$ |
| | TabDiff | $\mathbf{0.0001 \pm 0.0000}$ | $\mathbf{0.0007 \pm 0.0001}$ | $0.0049 \pm 0.0003$ |
| | SurvivalGAN | $0.0002 \pm 0.0001$ | $0.0045 \pm 0.0016$ | $0.0124 \pm 0.0033$ |
| | SurvDiff (*ours*) | $0.0004 \pm 0.0002$ | $0.0058 \pm 0.0016$ | $0.0075 \pm 0.0034$ |

Table 10: **KM metrics.** Kaplan-Meier metrics across multiple runs over different datasets (reported: mean ± s.d.) over 10 runs with different seeds.

# G    SENSITIVITY STUDY: REDUCED DATASET SIZES

We further investigate the performance of SURVDIFF under reduced dataset sizes by randomly downsampling the AIDS and METABRIC datasets. Table 11 summarizes the results in comparison to TabDiff across the (i) *covariate distribution fidelity* metrics, the (ii) *survival analysis performance* metrics, and (iii) *survival* metrics. Across most metrics and settings, SURVDIFF achieves clear improvements, with only three exceptions in which the results remain comparable. On all other metrics, SURVDIFF demonstrates superior performance. Notably, on METABRIC, the gains are substantial, with *large improvements* in Wasserstein distance, Brier score, RMST gap, and KM MSE. This is particularly relevant since METABRIC is the dataset where both methods were previously on par in the larger-scale evaluation. The results thus underscore that SURVDIFF not only retains its strength in smaller-sample regimes but, in fact, shows *even stronger advantages for smaller datasets*. ⇒ *These findings highlight the robustness of our approach when data availability is limited.*

| Metric | Method | AIDS (500) | AIDS (700) | METABRIC (500) | METABRIC (700) |
|---|---|---|---|---|---|
| **JS distance** | TabDiff | **0.0083 ± 0.0010** | **0.0086 ± 0.0006** | 0.0300 ± 0.0007 | 0.0280 ± 0.0008 |
| (↓: better) | SurvDiff (*ours*) | **0.0083 ± 0.0012** | 0.0092 ± 0.0007 | **0.0048 ± 0.0017** | **0.0031 ± 0.0005** |
| **Wasserstein distance** | TabDiff | 0.1801 ± 0.0432 | 0.1398 ± 0.0326 | 0.1066 ± 0.0332 | 0.0882 ± 0.0119 |
| (↓: better) | SurvDiff (*ours*) | **0.1280 ± 0.0079** | **0.1211 ± 0.0157** | **0.0877 ± 0.0069** | **0.0774 ± 0.0062** |
| **C-Index** | TabDiff | 0.6303 ± 0.0698 | 0.5818 ± 0.0428 | **0.6452 ± 0.0178** | 0.6275 ± 0.0282 |
| (↑: better) | SurvDiff (*ours*) | **0.7401 ± 0.0533** | **0.6482 ± 0.0268** | 0.6431 ± 0.0338 | **0.6343 ± 0.0475** |
| **Brier Score** | TabDiff | 0.0702 ± 0.0065 | 0.0872 ± 0.0031 | 0.1750 ± 0.0087 | 0.2025 ± 0.0067 |
| (↓: better) | SurvDiff (*ours*) | **0.0588 ± 0.0023** | **0.0840 ± 0.0013** | **0.1692 ± 0.0060** | **0.2006 ± 0.0017** |
| **RMST gap** | TabDiff | 0.0361 ± 0.0240 | 0.0235 ± 0.0168 | 0.0092 ± 0.0035 | 0.0184 ± 0.0171 |
| (↓: better) | SurvDiff (*ours*) | **0.0091 ± 0.0042** | **0.0119 ± 0.0062** | **0.0064 ± 0.0024** | **0.0120 ± 0.0043** |
| **KM MSE** | TabDiff | 0.0060 ± 0.0055 | 0.0029 ± 0.0029 | 0.0011 ± 0.0002 | 0.0026 ± 0.0019 |
| (↓: better) | SurvDiff (*ours*) | **0.0003 ± 0.0001** | **0.0006 ± 0.0005** | **0.0010 ± 0.0002** | **0.0019 ± 0.0004** |

Table 11: **Downsampled datasets.** Covariate fidelity, downstream performance, and survival metrics over different *downsampled* datasets (reported: mean ± s.d.) across 10 runs with different seeds.

# H   SURVDIFF TRAINING LOSS

The training losses are shown in Figure 15, which shows smooth and stable convergence across all objectives. Both the discrete and continuous diffusion losses decrease steadily, which reflects effective denoising of categorical and numerical covariates. The survival loss declines in parallel, indicating that the additional supervision integrates well with the generative process. Evidently, in the total loss, the *adaptive scaling* of $\lambda_{\text{surv}}$ balances the different components during training.

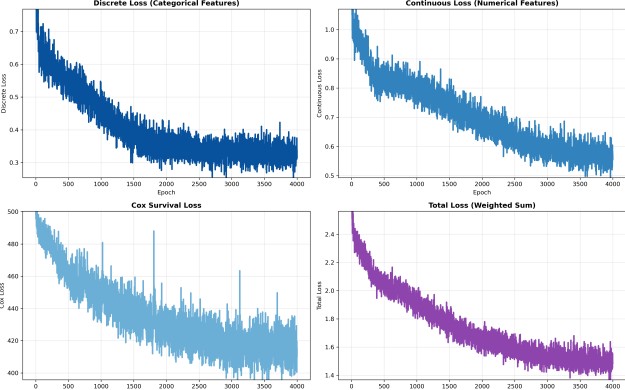

Figure 15: **Training dynamics of SURVDIFF.** Shown are the discrete/categorical diffusion loss $\mathcal{L}_{\text{disc}}$, the continuous diffusion loss $\mathcal{L}_{\text{cont}}$, the Cox survival loss $\mathcal{L}_{\text{surv}}$, and the total objective $\mathcal{L}_{\text{total}} = \mathcal{L}_{\text{diff}} + \lambda_{\text{surv}}\mathcal{L}_{\text{surv}}$.

## I  DIFFERENTIALLY-PRIVATE SURVDIFF

We further present a variant of SURVDIFF that is differentially private (Dwork & Roth, 2014). For this, we combine SURVDIFF with differentially private stochastic gradient descent (DP-SGD) (Abadi et al., 2016). Differential privacy (DP) provides a *formal guarantee* that the influence of any single individual in the training set is negligible. Formally, a randomized mechanism $M$ is $(\varepsilon, \delta)$-differentially private if for all adjacent datasets $D$ and $D'$ differing in one record, i.e.,

$$P[M(D) \in S] \ \leq \ e^{\varepsilon} P[M(D') \in S] \ + \ \delta \qquad \text{for all measurable } S. \tag{18}$$

This constraint enforces that the distribution of the model's output changes only minimally when a single patient is removed or replaced, thereby limiting what can be inferred about any individual (Abadi et al., 2016). Note that none of the baselines (i.e., differentially-private variants of both SurvivalGAN and Ashhad are lacking). To this end, our DP-SURVDIFF is the **first** *differentially-private* method for synthetic survival data generation.

DP-SGD ensures this guarantee by clipping per-sample gradients to a fixed norm $C$ and adding Gaussian noise scaled to the clipping threshold. At iteration $t$, the update is

$$g_t \ = \ \frac{1}{B} \sum_{i \in \mathcal{B}_t} \text{clip}(\nabla_\theta \ell_i, \, C) \ + \ \mathcal{N}\big(0, \, \sigma^2 C^2 I\big), \tag{19}$$

where $\sigma$ is the noise multiplier.

In all private experiments, we impose the same privacy budget for both DP-SURVDIFF and the DP-GAN baseline, fixing $\varepsilon = 8.0$ and $\delta = 10^{-5}$. This budget is in line with typical DP deep-learning practice and provides a meaningful privacy guarantee while maintaining a usable signal for model learning (Abadi et al., 2016).

Table 12 summarizes the results across two datasets and ten random seeds. Under identical privacy constraints, DP-SURVDIFF consistently achieves better C-Index, Brier Score, and divergence metrics compared to DP-GAN, indicating that our model remains robust even under strict privacy-preserving training.

| Metric | Method | AIDS | METABRIC |
|---|---|---|---|
| **C-Index** ($\uparrow$: better) | DP-GAN | $0.4872 \pm 0.0261$ | $0.4872 \pm 0.0261$ |
| | DP-SurvDiff (*ours*) | $\mathbf{0.5104 \pm 0.0222}$ | $\mathbf{0.5090 \pm 0.0144}$ |
| **Brier Score** ($\downarrow$: better) | DP-GAN | $0.4083 \pm 0.0304$ | $0.2595 \pm 0.0188$ |
| | DP-SurvDiff (*ours*) | $\mathbf{0.1298 \pm 0.0352}$ | $\mathbf{0.2461 \pm 0.0091}$ |
| **JS distance** ($\downarrow$: better) | DP-GAN | $0.1100 \pm 0.0053$ | $0.0525 \pm 0.0037$ |
| | DP-SurvDiff (*ours*) | $\mathbf{0.0570 \pm 0.0033}$ | $\mathbf{0.0365 \pm 0.0025}$ |
| **Wasserstein distance** ($\downarrow$: better) | DP-GAN | $2.1769 \pm 0.1217$ | $0.7631 \pm 0.0756$ |
| | DP-SurvDiff (*ours*) | $\mathbf{0.9654 \pm 0.0852}$ | $\mathbf{0.4135 \pm 0.042}$ |
| **Shape error rate** ($\downarrow$: better) | DP-GAN | $0.6075 \pm 0.0297$ | $0.3323 \pm 0.0240$ |
| | DP-SurvDiff (*ours*) | $\mathbf{0.3416 \pm 0.0270}$ | $\mathbf{0.2721 \pm 0.0187}$ |

Table 12: **Extension of SURVDIFF to differential privacy.** Metrics across multiple runs over different datasets (reported: mean $\pm$ s.d.) over 10 runs with different seeds.

## J  ABLATION STUDY AND PARAMETER SENSITIVITY ANALYSIS

We conduct an ablation study to isolate the contribution of the survival loss weighting mechanism used in SURVDIFF. In this variant, we fix the survival loss weight to $w = 1$, removing the down-weighting of sparse risk sets and treating all event times uniformly. Table 13 reports results across ten runs on GBSG2 and METABRIC. The full method achieves better C-Index and Brier Score and typically attains lower divergence metrics, with performance gains that are consistent across datasets.

The differences are moderate, as expected for a stable objective, but the pattern is systematic rather than incidental, indicating that duration-dependent weighting provides a measurable benefit without introducing variability or instability.

| Metric | Method | GBSG2 | METABRIC |
|---|---|---|---|
| **C-Index** ($\uparrow$: better) | $w = 1$ | $0.6545 \pm 0.0266$ | $0.5990 \pm 0.0206$ |
|  | $w^*$ | $\mathbf{0.6601 \pm 0.0252}$ | $\mathbf{0.5992 \pm 0.0272}$ |
| **Brier Score** ($\downarrow$: better) | $w = 1$ | $0.2041 \pm 0.0089$ | $0.2083 \pm 0.0066$ |
|  | $w^*$ | $\mathbf{0.2037 \pm 0.0092}$ | $\mathbf{0.2069 \pm 0.0071}$ |
| **JS distance** ($\downarrow$: better) | $w = 1$ | $0.0075 \pm 0.0008$ | $0.0067 \pm 0.0016$ |
|  | $w^*$ | $\mathbf{0.0074 \pm 0.0007}$ | $\mathbf{0.0062 \pm 0.0013}$ |
| **Wasserstein distance** ($\downarrow$: better) | $w = 1$ | $\mathbf{0.0344 \pm 0.0028}$ | $0.0554 \pm 0.0062$ |
|  | $w^*$ | $0.0347 \pm 0.0026$ | $\mathbf{0.0535 \pm 0.0059}$ |

Table 13: **Ablation study.** Metrics across multiple runs over different datasets (reported: mean $\pm$ s.d.) over 10 runs with different seeds.

We further study the sensitivity of SURVDIFF to the exponential-decay parameter $\alpha_{\mathrm{surv}}$, which moderates the contribution of long-duration events in the time-sensitive survival loss. Table 14 summarizes results for $\alpha_{\mathrm{surv}} \in \{0.01, 0.1, 0.15, 0.25\}$ on GBSG2 and METABRIC. Across all settings, SURVDIFF exhibits stable performance with only small variants in C-Index, Brier Score, JS distance and Wasserstein distance. The configuration $\alpha_{\mathrm{surv}} = 0.1$ yields consistently strong results on both datasets. These findings show that SURVDIFF maintains robustness over a reasonable range of $\alpha_{\mathrm{surv}}$ values, supporting its practical applicability without requiring extensive hyperparameter tuning.

| Metric | Method | GBSG2 | METABRIC |
|---|---|---|---|
| **C-Index** ($\uparrow$: better) | $\alpha = 0.01$ | $0.6598 \pm 0.0272$ | $0.5933 \pm 0.0332$ |
|  | $\alpha = 0.15$ | $0.6519 \pm 0.0318$ | $0.5934 \pm 0.0313$ |
|  | $\alpha = 0.25$ | $0.6561 \pm 0.0273$ | $0.5989 \pm 0.0253$ |
|  | $\alpha = 0.1$ | $\mathbf{0.6601 \pm 0.0252}$ | $\mathbf{0.5992 \pm 0.0272}$ |
| **Brier Score** ($\downarrow$: better) | $\alpha = 0.01$ | $0.2039 \pm 0.0094$ | $0.2083 \pm 0.0067$ |
|  | $\alpha = 0.15$ | $0.2039 \pm 0.0092$ | $0.2109 \pm 0.0069$ |
|  | $\alpha = 0.25$ | $0.2042 \pm 0.0108$ | $0.2087 \pm 0.0063$ |
|  | $\alpha = 0.1$ | $\mathbf{0.2037 \pm 0.0092}$ | $\mathbf{0.2069 \pm 0.0071}$ |
| **JS distance** ($\downarrow$: better) | $\alpha = 0.01$ | $\mathbf{0.0071 \pm 0.0009}$ | $0.0070 \pm 0.0015$ |
|  | $\alpha = 0.15$ | $0.0081 \pm 0.0008$ | $0.0076 \pm 0.0014$ |
|  | $\alpha = 0.25$ | $0.0077 \pm 0.0008$ | $0.0067 \pm 0.0018$ |
|  | $\alpha = 0.1$ | $0.0074 \pm 0.0007$ | $\mathbf{0.0062 \pm 0.0013}$ |
| **Wasserstein distance** ($\downarrow$: better) | $\alpha = 0.01$ | $0.0349 \pm 0.0028$ | $0.0556 \pm 0.0057$ |
|  | $\alpha = 0.15$ | $0.0364 \pm 0.0025$ | $0.0598 \pm 0.0059$ |
|  | $\alpha = 0.25$ | $0.0349 \pm 0.0029$ | $0.0559 \pm 0.006$ |
|  | $\alpha = 0.1$ | $\mathbf{0.0347 \pm 0.0026}$ | $\mathbf{0.0535 \pm 0.0059}$ |

Table 14: **Sensitivity analysis.** Metrics across multiple runs over different datasets (reported: mean $\pm$ s.d.) over 10 runs with different seeds.

