# OpenReview forum: "SurvDiff: A Diffusion Model for Generating Synthetic Data in Survival Analysis"
_ICLR.cc/2026/Conference — Submitted to ICLR 2026_

### Official Review · Reviewer_ctHa · 2025-10-31

**Soundness:** 1
**Presentation:** 3
**Contribution:** 2
**Rating:** 2
**Confidence:** 4

**Summary:**

This paper aims to build a generative model that is specifically tailored for survival analysis data generation, which is a known application problem. The paper introduced SurDiff model to address the problem, which is an end-to-end diffusion model that can generate both continuous and discrete variables at the same time. During training, the model is guided by a survival loss function to match the true survival event distribution across samples. Experiments show that SurvDiff can reproduce realistic event-time distribution and is potentially useful for downstream application. The authors claim that existing methods are not able to reproduce realistic event-time distributions and preserve censoring mechanisms, while SurvDiff can success in these two aspects with the help of survival loss.

**Strengths:**

1. Survival analysis is one of the critical problems in the medical area. The proposed method SurvDiff contains a survival loss that is specifically tailored for addressing the problem, making original contribution to the area.
2. SurvDiff is the first diffusion-based survival data generation model with survival task-specific design. Compared with SurvivalGAN, it is an end-to-end method that tends to avoid error propagation and is able to be trained stably.
3. The presentation is clear and well-structured, making the paper easy to follow. The choices of evaluation metrics, baseline methods and datasets are comprehensive.

**Weaknesses:**

1. Covariate distribution experiments: Missing implementation details, e.g. the number of samples used to compute these metrics, how is each covariate vector normalized for dimension reduction computation, how many bins are used to estimate probability density, how anomaly values are addressed.
2. Numerical results mismatch between JS distance and Wasserstein distance metrics. More explanations on this phenomenon are expected to help understand the true performance of the proposed method.
3. Downsampled dataset settings are not sufficiently justified. Does this setting mean that the dataset used to trained SurvDiff and all baselines are downsampled? What is the downsampling ratio and how will the performance change with varying downsampling ratio (including the full-dataset scenario)? From my perspective, it is still meaningful testing on full dataset to show the performance of SurvDiff.
4. Survival metrics are weak, which raises concerns on the effectiveness of the survival specific loss. TabDiff turns out to be the best baseline, which should just be equivalent to remove the survival loss from SurvDiff. Since the survival loss stands for the core contribution of this paper, these results make the contribution of this paper questionable.
5. Missing ablation studies and parameter sensitivity analysis. The effectiveness of survival loss, sparsity-aware weighting, and the loss weight parameters in SurvDiff are not fully explored. For downstream tasks, it is also critical to evaluate the downstream model performance without using any synthetic data generation methods to demonstrate benefits.

The major problem of this submission is that the claimed contribution is not sufficiently justified by the experiments. In fact, some experiments are showing opposite results: survival loss doesn’t really help survival task, while the naive TabDiff can perform notably better than SurvDiff. Without proper justifications, I believe this paper is not ready for publication yet, and major revisions are needed on both the methodology and experiment.

**Questions:**

I have some questions on parts where I didn’t fully understand: In eq.(10-11), the weights are not normalized. When trying to balance two loss components with eq.(13), how will the number of samples $n$ (for calculating eq.(10)) influence the optimal target fraction $\alpha_{\text{surv}}$? How is this $n$ selected? What is the definition of $\tau$? Is this $\mathcal{L}_{\text{surv}}$ calculated across the batch size dimension?

I have another minor suggestion for the authors: The two colors in Figure 3-8 are too similar to visually show the discrepancy between distributions of real and synthetic data. It would be better to opt for more contrasting colors.

---

> ### Author Response · Authors · 2025-11-20
>
> Thank you for the constructive feedback. We followed your feedback closely and have incorporated the suggested clarifications and additions in the revised manuscript (see **revised PDF**). Below, we address each point and explain how we improved our manuscript as a result..
>
> # Response to Weaknesses
>
> >**W1 – Clarification of the evaluation setup.**
>
> Thanks for pointing this out. Our revised manuscript now includes a **detailed** description of the covariate distribution experiments (see our **new Appendix B.1**). For each dataset with $n$ training samples and $p$ covariates, we generate exactly $n$ synthetic samples per method, apply the same preprocessing, and report the marginal Jensen-Shannon distance and the joint Wasserstein distance. In Appendix B.1, we further specify the histogram construction, the use of SciPy’s \[5\] JSD implementation, and the Sinkhorn-regularized 2-Wasserstein distance with feature-wise min-max scaling, matching the configuration of the SynthCity \[6\] metrics used for our baselines TabDiff and SurvivalGAN. SynthCity is a widely used library for tabular data synthesis and evaluation that provides common baselines and metric implementations. We also report detailed hyperparameter grids (which we had already included in the original submission). Please let us know if you think that any aspect of the evaluation requires further explanation.
>
> **Action**: We added **new Appendix B.1** with full implementation details including the computation of synthetic samples and metrics.
> \[5\] [https://github.com/vanderschaarlab/synthcity](https://github.com/vanderschaarlab/synthcity)
> \[6\] [https://scipy.org/](https://scipy.org/)
>
> >**W2 – Interpretation of JSD vs. Wasserstein**
>
> Thank you. In the revised manuscript, we now clarify that JSD is reported as a marginal feature-wise divergence, while Wasserstein is computed as a joint multivariate optimal-transport distance on the full covariate vectors. Hence, both metrics quantify different properties of the distributions, so differing rankings are expected rather than being indicative of error. Importantly, by choosing this setup, our work closely aligns with the configurations used in SynthCity \[1\] and in the SurvivalGAN/TabDiff benchmarks.
>
> **Action**: We clarified that we used marginal vs. joint metrics (see **Section 6** and our **new Appendix B.1**).
>
> >**W3 – Downsampled dataset clarification**
>
> Thank you for allowing us to clarify this: **all main results and conclusions in the paper are based on models trained on the full datasets for all methods.** Hence, all performance comparisons are comprehensive and fair. The downsampled settings appear only in the appendix as an additional robustness study, where we downsample uniformly at random and train on the same reduced set.
>
> >**W4 – Clarification of survival-related metrics**
>
> Thank you for this suggestion, based on which we clarified our manuscript as follows:
>
> 1. In the revised manuscript, we  renamed the Kaplan-Meier-based quantities from “survival metrics” to “KM metrics” and state explicitly that they assess the population-level Kaplan-Meier curve.
> 2. We introduced a broader set of survival-related evaluations. In addition to KM metrics, we now report **new survival distribution** metrics (an event-time-distribution shape score that includes T and E in **new** **Appendix C** and **updated Table 2**), marginal histograms for X, T, E (**new** **Figures 7-9**), and temporal histograms for event and censoring counts (**new**  **Figures 4, 13-14**). We now explicitly refer to downstream evaluations as “survival analysis performance” to make their role and connection to the survival task clearer.
>
> With this extended evaluation, we confirm the **strong benefits of SurvDiff** in faithfully capturing both event-time and censoring mechanisms. Importantly, **SurvDiff is superior to TabDiff across three dimensions**: (1) covariate distribution metrics, (2) survival distribution metrics, and (3) downstream survival model performance. **Often, SurvDiff is better by a substantial margin, e.g.** 36-58% for Jensen-Shannon Distance (see Table 2). The *only* dimension where both are comparable over all datasets is the KM-based metric. Together, these results demonstrate that SurvDiff not only maintains strong covariate realism but also better captures the underlying survival and censoring dynamics, thereby addressing key weaknesses of existing tabular generators and achieving state-of-the-art performance for generating synthetic survival data.
>
> **Action**: We added **new survival distribution** metrics and added new comparisons (see **Appendix C**). This confirms that our proposed SurvDiff method is better in preserving the time-to-event mechanism by a large margin.

---

> > ### Author Response · Authors · 2025-11-20
> >
> > >**W5 – Ablations, parameter sensitivity, and real-data baseline**
> >
> > Thanks for the suggestion. We now include a **new** ablation study that isolates the contribution of the survival-specific loss component and a **new** parameter sensitivity analysis (see **new Appendix J**).
> >
> > To address the reviewer’s comment on downstream performance, we additionally include a **new real-data baseline** where we train the downstream survival model **directly on the real training data without any synthetic augmentation** (see **updated Table 3**).
> >
> > **Action**: We added **new** ablations and sensitivity analyses (see **new Appendix J**) and a **new** real-data baseline in **Table 3**.
> >
> > # Response to Questions
> >
> > >**Q1a – Weights in Eq. (10)-(11)**
> >
> > Thank you for allowing us to elaborate. The weights in Eq. (10)-(11) are *importance weights*, **not** probabilities. They are only used to downweight very late events in the likelihood and are not meant to form a normalized weighted average. Normalizing them is neither required nor intended, only relative magnitudes matter. Multiplying all weights by a constant would simply be absorbed into $\\lambda\_{\\text{surv}}$.
> >
> > **Action:** We added a clarification of the role of the weights (see Section 4.3).
> >
> > >**Q1b – The role of $n$, $\\alpha{\\text{surv}}$, and the calibration**
> >
> > In Eq. (10) we use $n$ to denote the minibatch size, and $\\mathcal{L}\_{\\text{surv}}$ is implemented exactly as written there: a sum over all uncensored samples in the current batch, i.e., $\\mathcal{L}\_{\\text{surv}}$ indeed grows with the number of events in the batch. Equation (13) uses running averages of the summed losses over a short calibration window. $\\lambda\_{\\text{surv}}$ is chosen so that, up to the small $\\varepsilon$, the survival term contributes a target fraction $\\alpha{\\text{surv}}$ of the total objective, irrespective of the common scale of the diffusion and the survival loss terms.
> >
> > **Action:** We added a short intuition on the role of the above variables (see Section 4.3).
> >
> > >**Q2 – Definition of $\\tau$**
> >
> > $\\tau$ is set to the $q$-quantile of the event times in the training set, which defines the cutoff above which the exponential downweighting starts.
> >
> > **Action:** We **now** state this choice explicitly in Section 4.3.
> >
> > Additionally, we have **updated the color scheme** of the figures to use a more contrasting color pair for real vs. synthetic data to improve readability.
> >
> > We thank the reviewer again for the constructive input. In summary, we now provide **all**  implementation details, **new** ablations and sensitivity studies, and **improved** structure of the survival-related evaluations to strengthen our work. We are confident that our paper has improved substantially as a result.

---

### Official Review · Reviewer_Tnii · 2025-11-01

**Soundness:** 3
**Presentation:** 3
**Contribution:** 2
**Rating:** 4
**Confidence:** 3

**Summary:**

The paper presents SurvDiff, an end-to-end diffusion model specifically designed for generating synthetic survival data. The key contribution is jointly modeling covariates, event times, and censoring mechanisms through a survival-tailored loss function.

**Strengths:**

1. Well-motivated problem: The paper addresses a need in medical research where survival data with censoring mechanisms presents unique challenges for synthetic data generation and downstream applications.
2. Comprehensive experiments: The evaluation across AIDS, GBSG2, and METABRIC datasets is thorough, showing consistent improvements over baselines in both covariate fidelity and downstream performance.
3. Clear presentation: The paper is generally well-written with good visualizations (t-SNE plots, KM curves) that effectively demonstrate the method's performance.

**Weaknesses:**

*Limited Novelty*
1. The core diffusion framework closely follows TabDiff (Shi et al., 2024b) for mixed-type data
2. The main novel contribution is adding a weighted Cox loss (Equation 10) with exponential decay weighting (Equation 11)
.The combination of existing techniques (masked diffusion for discrete, Gaussian for continuous) is straightforward

*Lack of Theoretical Guarantees*
1. No theoretical justification for why the proposed loss preserves censoring mechanisms
2. Missing bounds on the quality of generated distributions

*Missing Privacy Analysis* The paper lack privacy analysis, such as membership inference attack evaluation. No discussion of potential patient re-identification risks

**Questions:**

Only compares against SurvivalGAN as a survival-specific method. Could benefit from comparisons with other recent tabular generative models.

---

> ### Author Response · Authors · 2025-11-20
>
> Thank you for the constructive feedback. We followed your feedback closely and have incorporated the suggested clarifications and additions in the revised manuscript (see **revised PDF**). Below, we address each point and explain how we improved our manuscript as a result.
>
> # Response to Weaknesses
>
> >**W1 – Novelty**
>
> Thanks for pointing this out. While our backbone follows TabDiff, we make several important changes to improve the performance for our specific task. **Our key contribution is in adapting this backbone to handle censoringing structure and survival risk dynamics during training.** This requires a *survival-specific objective that interacts with the diffusion process in ways that do not arise in standard tabular generation*. The survival head must learn risk-set structure and proportional hazards behaviour while the backbone denoises mixed-type covariates, which changes the training dynamics relative to TabDiff. The weighting scheme and calibration mechanism are introduced to maintain stability when risk sets shrink and gradients become high-variance, a challenge unique to survival modelling. Recent diffusion-based tabular methods do not address survival outcomes, and recent survival-related diffusion work does not perform joint diffusion over (X,T,E). As a result, we show also empirically that SurvDiff outperforms TabDiff by a clear margin (e.g., 36-58% for Jensen-Shannon Distance and 13-32% for Event-Time Discrepancy).
>
> >**W2 – Theoretical Guarantees/ Bounds**
>
> To the best of our knowledge, such theoretical guarantees or bounds are not provided for the baselines we study. Existing generative approaches for synthetic tabular or survival data, particularly diffusion-based models that currently achieve state-of-the-art performance, are typically evaluated empirically rather than through provable distributional error bounds. Should the reviewer have specific literature in mind where such guarantees are established, we would be happy to review and incorporate it.
>
> Precisely because such guarantees are not available in this line of work, we put substantial effort into a broad empirical assessment. Beyond standard covariate metrics (JS/ Wasserstein on X), survival model performance, and Kaplan Meier metrics, we **now added new** (i) temporal distribution plots of censored and uncensored events (**Figures 4, 13-14**), (ii) the Event-Time Divergence metric (**Appendix C**), which compares real and synthetic covariates of uncensored individuals within matched time intervals, (iii) the Shape metric, which includes T and E in the marginal comparison (**updated Table 2**), and (iv) normalized marginal histograms for X, T, and E (**Figures 7-9**).
>
> Together, these evaluations provide a detailed empirical picture of event-time realism, censoring mechanics, and covariate-time structure.
>
> **Action**: We added **new evaluation metrics and visualizations** **(updated Table 4, new Figures 4, 7-9, 13-14, and new Appendix C).**
>
> >**W3 – Extension to Privacy Analysis**
>
> Thank you for this suggestion. To support downstream use cases where privacy matters, **we demonstrate how our SurvDiff method can be extended to incorporate differential privacy (DP)**. For this, we present a **new variant called “DP-SurvDiff”.** Therein, we combine SurvDiff with DP-SDG to ensure differential privacy, therefore attacks such as membership-inference attacks are prevented. We also demonstrate our new method called DP-SurvDiff across various numerical experiments (see **new Appendix I**). Therein, we compare DP-SurvDiff against the only known DP-baseline for survival data (DP-GAN). Overall, we find that the synthetic data of SurvDiff is substantially higher fidelity.
>
> **Action**: We **added a new differentially private variant of our method called “DP-SurvDiff”.** We further demonstrate the performance of DP-SurvDiff across numerical experiments (see **new Appendix I**).

---

> > ### Author Response · Authors · 2025-11-20
> >
> > # Response to Questions
> >
> > >**Q1 – Baseline Coverage**
> >
> > We appreciate the reviewer’s point regarding the breadth of baselines and agree that comparative coverage is important for assessing generative models. To clarify, the original submission already included several recent and widely used tabular generators: **CTGAN**, **TVAE**, **TabDiff**, and **NFlows** in addition to SurvivalGAN (see Tables 2 and 3). These general-purpose models are strong generators and represent the dominant baselines in current tabular synthesis research. **This establishes that SurvDiff is superior to existing survival-specific methods and to the strongest general tabular baselines (i.e., CTGAN**, **TVAE**, **TabDiff**, **and NFlows)**.
> >
> > Additionally, we **now also added a recent, survival-specific method** (named “Ashhad” \[1\]). The baseline was published at MLHC only *after the ICLR Deadline*. Importantly, we show that our method outperforms \[1\] by a clear margin.
> >
> > **Action**: We **added a recent survival-related baseline** \[1\] (see Tables 2 and 3). We refer to \[1\] as “Ashhad”. Importantly, the baseline \[1\] was published at MLHC only after the ICLR deadline.
> >
> > We thank the reviewer for the thoughtful comments across the entire review. The revision above strengthens the clarity, completeness, and transparency of the manuscript.
> >
> > \[1\] Ashhad M, Henao R. Generating Accurate Synthetic Survival Data by Conditioning on Outcomes. In Machine Learning for Healthcare Conference 2025 Oct 7. PMLR.

---

### Official Review · Reviewer_biW2 · 2025-11-01

**Soundness:** 3
**Presentation:** 1
**Contribution:** 3
**Rating:** 4
**Confidence:** 3

**Summary:**

The paper proposes a diffusion model for generating synthetic data for survival analysis. It achieves this by introducing a survival analysis specific loss function. The resulting architecture is evaluated across multiple datasets, focusing on 1) covariate distribution similarity, 2) downstream task performance, 3) survival outcome reproduction and 4) robustness in low-data regimes. The model shows strong performance on 1, 2 and 4 while showing comparable performance on 3.

**Strengths:**

- Strong performance on both covariate distribution fidelity and downstream tasks.
- Addresses an important problem, synthetic data for healthcare applications.

**Weaknesses:**

1. Weak connection between the drawbacks of existing works and the chosen evaluation. The introduction states that existing general-purpose methods for generating tabular data fail to: 1) reproduce realistic event-time distributions and 2) preserve censoring mechanics. The evaluation is based on comparing covariate distributions, not event-time distributions, as well as downstream tasks. It is unclear whether the downstream tasks evaluate both event-time distribution realism and censoring mechanics preservation.
2. The clarity of the paper could benefit from additional intuition when introducing key concepts. While these might be common in the domain of survival analysis, they might not be familiar to the general ICLR audience.
	1. (Right-)Censoring: First used in introduction, definition and intuition given in 3.1
	2. Time-to-event: First used in introduction, definition and intuition given in 3.1
	3. Risk sets: Defined in 4.3, no intuition given
3. The notation in section 4.3 was challenging and difficult to follow.
4. No discussion around memorization, there is always a risk of the model learning to reproduce individuals from the training data, which could raise privacy concerns.

**Questions:**

1. Are the event indicator and event time variables part of the covariate distribution fidelity study?
2. How are the different evaluations related to the stated goals of the paper?
	1. reproduce realistic event-time distributions
	2. preserve censoring mechanics.
3. Could you elaborate on how $\mathcal{L}\_\text{surv}$ is calculated? The survival head $f_\theta$ takes the continuous covariates and the discrete covariate probabilities, both denoised by the reverse diffusion process, and predicts a scalar risk score $r_i$ for a specific individual $i$. This $i$ refers to a synthetic sample. In equation 10, it is then compared to other individuals using the risk set in the denominator. Are these individuals also synthetic or are they drawn from the training data?
4. Questions to clarify the notation in section 4.3:
	1. In equation 10, is the denominator supposed to be $\exp(r_j)$ rather than $\exp(r_i)$?
	2. In equation 11, what are the definitions of $\tau$ and $T_i$? How are they related to $T$ and $t_i$?

---

> ### Author Response · Authors · 2025-11-20
>
> Thank you for the constructive feedback. We followed your feedback closely and have incorporated the suggested clarifications and additions in the revised manuscript (see **revised PDF**). Below, we address each point and explain how we improved our manuscript as a result.
>
> # Response to Weaknesses
>
> >**W1 – Evaluations of event-time and censoring realism**
>
> Thank you for this helpful comment. We appreciate the opportunity to more closely align our evaluation with the identified gaps in the literature and the stated objectives of our method (=preserving the event-time mechanism). To address this, we revised our evaluation by adding **new analyses that explicitly assess event-time fidelity and the realism of censoring in the generated data** (see new Figures 4, 13-14, Tables 7-9, and Appendix E). Specifically, we improved our manuscript in the following ways:
>
> 1. **New visualization of the event-time fidelity.** We added **new temporal distribution plots** showing the occurrence of censored and uncensored events over time (see our **new Figures 4 and new Appendix E**). These directly compare how real and synthetic patients experience events and censoring across the follow-up horizon. Therein, we clearly see that *our proposed method (=SurvDiff) is strong in preserving the event-time fidelity*.
> 2. We add **new normalized marginal histograms** for the covariates, event-time, and event-indicator (**new Figures 13-14**) to provide a direct visual comparison. Therein, we again confirm that SurvDiff provides the most faithful reconstruction across variables.
> 3. We **now include the Shape metric** from TabDiff \[4\], which compares marginal distributions one variable at a time, including event-time and event-indicator variables (see **new result in Table 2**). We report that SurvDiff provides the best overall scores among the evaluated models.
> 4. We introduce a **new Event-Time Divergence (ETD)** metric (**new Appendix C**). The idea behind ETD is to directly quantify the fidelity in preserving the event-time distributions. For this, ETD partitions the event-time axis into equal-mass intervals and computes the Jensen-Shannon distance between real and synthetic covariates for uncensored individuals. This evaluates whether **patients who die/experience an event at similar times share similar characteristics** across real and synthetic data. *As a result, we find SurvDiff outperforms over all datasets.*
>
>
> The new evaluations confirm that our method is overall best in capturing the realism of both event-times and censoring mechanisms. In contrast, existing methods show varying levels of mismatch in these event-time dimensions. Hence, our SurvDiff method demonstrates superior performance in preserving the event-time dimension.
>
> **Action**: We **added new evaluations (temporal plots, Event-TIme-Divergence (ETD), Shape, and histograms) to numerically and visually demonstrate the drawbacks of existing methods in capturing event-time fidelity and further show the benefit of ours**. For this, among others, we introduce **a new metric** called Event-Time Divergence (ETD) that directly quantifies how well the targeting event-time fidelity are preserved (**new Appendix C**)**.**
>
> >**W2 – Added intuitive explanations**
>
> Thank you for the suggestion. We revised our manuscript to introduce these concepts (i.e., censoring, time-to-event, etc.) in the introduction. We hope this improves the accessibility to readers unfamiliar with survival analysis.
>
> **Action**: We added short intuitive explanations for key survival-analysis concepts (see **revised Introduction**).
>
> >**W3 – Improved unified notation**
>
> Thank you for pointing this out. The distinction between $T\_i$ and $t\_i$ was unintended. Upon reading your comment, we realized that we should clarify the notation: we **now unified the notation** in Section 4.3, and clarified the definition of $\\tau$ in the time-weighted loss.
>
> **Action**: We **improved the notation** for $T\_i$ and $t\_i$, and added a concise definition of $\\tau$.
>
> >**W4 – Added differential privacy experiment**
>
> We demonstrate how our SurvDiff method can be extended to incorporate differential privacy (DP). For this, we present a **new variant called “DP-SurvDiff”.** Therein, we combine SurvDiff with DP-SDG to ensure differential privacy. We also demonstrate our new method called DP-SurvDiff across various numerical experiments (see **new Appendix I**). Therein, we compare DP-SurvDiff to the DP-baseline DP-GAN. Overall, we find that the synthetic data of SurvDiff has substantially higher fidelity.
>
> **Action**: We **added a new differentially private variant of our method called “DP-SurvDiff”.** We further demonstrate the performance of DP-SurvDiff across numerical experiments (see **new Appendix I**).

---

> > ### Author Response · Authors · 2025-11-20
> >
> > # Response to Questions
> >
> > >**Q1 & Q2 Expanded evaluation of T and E**
> >
> > Thank you for the question. Yes, our fidelity studies now carefully assess both the event indicator (“E”) and the event time (“T”). In response to your comments, we re-organized our evaluation metrics (and now present also results for new event-time fidelity metrics) as follows:
> >
> > * **General data generation metrics**:
> > * **Covariate fidelity (“X”):** For covariate fidelity metrics, we follow standard practice and only evaluate X using the JS and Wasserstein distance.
> > * **Event-time fidelity**: In response to your comment, we **now include** (i) **new temporal distribution plots** for censored and uncensored events (see our **new Figures 4, 13-14)**, (ii) the **new Event-Time Divergence metric** which compares covariates of patients dying in similar time intervals (see our **new Appendix C**).
> > * **Overall fidelity analysis (“X”, “T”, “E”):** For fidelity studies including T and E, but also the covariates, we report (i) the **Shape metric** described above, which also includes T and E in the marginal comparison (see our updated **Table 2**), and (ii) the **normalized marginal histograms** for X, T, E (see our **new Figures 7-9**).
> > * **Survival analysis performance:** For survival model downstream performance, we report the C-Index and Brier score averaged on five different survival models (see Table 3).
> >
> > Both, the reproduction of realistic event-time distributions and the preservation of censoring mechanics are evaluated with the event-time and overall fidelity metrics. Covariate fidelity and survival analysis performance are also central components of synthetic data evaluation. Assessing the covariate distribution similarity and training standard survival models on synthetic data is well established practice and provides complementary evidence for the overall realism of generated data.
> >
> > **Action**: **We added new evaluations to** explicitly analyze the event-time censoring distributions (see our **new Figures 4, 13-14** and our **new Appendix C**) as well as evaluations that include our event-time censoring distributions (see our **new Figures 7-9** and our **updated Table 2**). We further re-organized the main paper (e.g., see Sec 3.2 and 6\) and added a concise summary in Sec 6, organizing the evaluation metrics by dimension, including a new survival-specific fidelity component.
> >
> > >**Q3 – Clarification of loss computation**
> >
> > Thank you for your question. We appreciate the opportunity to clarify this aspect. $\\mathcal{L}\_\\text{surv}$ is only applied to synthetic samples. The survival head takes denoised synthetic covariates and outputs a risk score $r\_i$ for each synthetic individual $i$. In the denominator of the partial likelihood, the risk set contains other synthetic individuals in the same batch. Importantly, no real samples appear in the loss, which *avoids leakage* from real to synthetic data during training.
> >
> > >**Q4 – Corrected equation and clarified definitions**
> >
> > 1. Thank you. You are correct: the denominator should involve rj. Thank you for the careful reading. We have now corrected this.
> > 2. Thank you for the suggestion. Upon reading your comment, we realized that we can unify the notation. We now use: $T\_i$: observed time (event or censoring) for the synthetic sample $i$
> > $t\_i$: same quantity as $T\_i$; notation is now unified in the revision, thank you for making us aware of this.
> > $T \= (T\_1,\\ldots , T\_n)$: vector of observed times for all samples in the batch
> > $\\tau$ \= duration threshold (e.g., 80th percentile of observed times) from which exponential downweighting starts.
> >
> > **Action**: We corrected the notation and added a concise definition of $\\tau$ (see our revisions to **Section 4.3**).
> >
> > Thank you again for the thoughtful and constructive feedback. We appreciate the suggestions, which we followed closely to revise our paper. We are confident that our paper has improved substantially as a result.
> >
> > \[4\] Juntong Shi, Minkai Xu, Harper Hua, Hengrui Zhang, Stefano Ermon, and Jure Leskovec. TabDiff: A mixed-type diffusion model for tabular data generation. In International Conference on Learning Representations (ICLR), 2024.

---

### Official Review · Reviewer_Ser3 · 2025-11-01

**Soundness:** 3
**Presentation:** 3
**Contribution:** 2
**Rating:** 2
**Confidence:** 4

**Summary:**

The authors propose DiffSurv, an end-to-end diffusion model for generating synthetic survival data. Unlike existing approaches, it jointly generates covariates, event times and censoring indicators. They present experimental results demonstrating the superiority of their approach relative to SurvivalGAN.

**Strengths:**

The proposed objective is the most innovative aspect of the proposed model. Specifically, they take a standard partial likelihood loss and add weights to mitigate the effects caused by the shrinking risk set as t goes to infinity.

**Weaknesses:**

The authors argue that there is only one method (SurvivalGAN) to generate synthetic survival data, however, there is [1], which was more recently presented at MLHC [2]. In fact in [2], they also consider a diffusion model (TabDDPM) as their backbone and as an unconditional sampler (X, T and E are sampled jointly), thus the presented approach is not even the first to use diffusion models.

The experimental results in Tables 2 and 3 indicate that the proposed model is comparable or better than SurvivalGAN, however, they do not seem better than those in [2].

[1] Ashhad M, Henao R. Conditioning on time is all you need for synthetic survival data generation. arXiv preprint arXiv:2405.17333. 2024 May 27.
[2] Ashhad M, Henao R. Generating Accurate Synthetic Survival Data by Conditioning on Outcomes. InMachine Learning for Healthcare Conference 2025 Oct 7. PMLR.

**Questions:**

- Consider acknowledging TabDDPM [3] in the related work.
- Is TabDiff as implemented in the experiments equivalent to optimizing (12) with lambda=0? if, not an ablation study will be important to illustrate the benefit of the proposed objective.
- An ablation study with w=1 vs. (11) is necessary to demonstrate the impact of the proposed weighting scheme on performance.

[3] Kotelnikov A, Baranchuk D, Rubachev I, Babenko A. Tabddpm: Modelling tabular data with diffusion models. InInternational conference on machine learning 2023 Jul 3 (pp. 17564-17579). PMLR.

---

> ### Author Response · Authors · 2025-11-20
>
> Thank you for the constructive feedback. We followed your feedback closely and have incorporated the suggested clarifications and additions in the revised manuscript (see **revised PDF**). Below, we address each point and explain how we improved our manuscript as a result.
>
> # Response to Weaknesses
>
> >**W1 – Related work and novelty**
>
> Thank you for pointing us to the references. We carefully added both \[1\] and \[2\] to our revised paper. Paper \[2\] was published only **after the ICLR submission deadline**, because of which it was not included in the original submission. We now discuss the differences from our work (e.g., see **updated Table 1**) and thereby spell out how our method is novel.
>
> In particular, we now include the above works as a **new benchmark** and compare against our method (see the **new results in Tables 2 and 3**). Note that \[2\] is the extension of \[1\], and so we listed both methods in a single row (“Ashhad”). Importantly, *we find that our method is superior by a clear margin*.
>
> The above finding can be expected: The framework in \[1\] and \[2\] does **not** *jointly* generate (X,T,E) but instead follows a staged procedure, where covariates are generated conditionally and event-times plus indicators are modeled separately. The term “unconditional” in \[1\] and \[2\] refers to their benchmarking setup, in which several generative models are compared in a joint generation setting. We also provide results for such unconditional generative models without survival-specific adaptations; e.g. see our results in Tables 2 and 3\. Therein, we clarify the distinction between their mechanism and ours. Hence, the main conceptual difference is that our framework generates (X,T,E) jointly through a *novel end-to-end loss*.
>
> **Action:** We added the suggested references to our paper (see our revised Introduction and Related Work). We further **added \[1\] and \[2\] as another baseline** to our experiments (see our **new results in Tables 2 and 3).**
>
> >**W2 – Comparison to results in \[2\]**
>
> Thank you for pointing us to the baseline in \[2\]. We **added a new comparison to the method** in \[2\], which we refer to as “Ashhad”. Note that \[2\] was published only after the ICLR deadline. The new results are in our **updated Tables 2 and 3**. We find that **SurvDiff outperforms \[2\], across all datasets**.
>
> Thank you also for the opportunity to clarify why our results are different from \[2\]. The experimental setup used in \[2\] and in our paper are not comparable. The authors in \[2\] report performance for the **best-performing generative model** within their framework, while we do not perform model selection. To enable fairer alignment, we instead compare their method with the same experimental setup and find that our method outperforms theirs across all datasets.
>
> **Action**: We added the new Ashhad benchmark to our experiments (see **new results in Tables 2 and 3**).

---

> > ### Author Response · Authors · 2025-11-20
> >
> > # Response to Questions
> >
> > > **Q1 – Related work**
> >
> > Thank you for the suggestion. We followed your suggestion closely and updated our Section 2 with related work accordingly
> >
> > **Action**: We have now **revised the Related Work section** to include TabDDPM \[3\] and clarified its relation to TabDiff (see **changes in Section 2**).
> >
> > > **Q2 – Clarification of the TabDiff ablation study**
> >
> > Yes, you are correct. Regarding TabDiff, we confirm that TabDiff corresponds to the diffusion objective without the survival-loss component in our experiments. This case is therefore already covered in the original results, and no additional ablation study is required. Instead, we include a new ablation (fixing $w=1$) and a parameter sensitivity analysis in our **new Appendix J**, which further demonstrates the consistency in our design choices.
> >
> > > **Q3 – Additional ablation study**
> >
> > We have added **a new ablation study** comparing SurvDiff to a variant with a fixed $w=1$, which removes the downweighting of late events. Across all datasets, the ablation yields consistent results, supporting the benefit of the proposed weighting scheme.
> >
> > **Action:** We have added the new ablation in our **new Appendix J** (see our new **Table 13**).
> >
> > Thank you again for the thoughtful and constructive feedback. We appreciate the suggestions, which we followed closely to revise our paper. We are confident that our paper has improved substantially as a result, and *we thereby confirm that our proposed method (“SurvDiff”) achieves state-of-the-art performance.*
> >
> > \[1\] Ashhad M, Henao R. Conditioning on time is all you need for synthetic survival data generation. arXiv preprint arXiv:2405.17333. 2024 May 27\.
> > \[2\] Ashhad M, Henao R. Generating Accurate Synthetic Survival Data by Conditioning on Outcomes. In Machine Learning for Healthcare Conference 2025 Oct 7\. PMLR.
> > \[3\] Kotelnikov A, Baranchuk D, Rubachev I, Babenko A. Tabddpm: Modelling tabular data with diffusion models. In International conference on machine learning 2023 Jul 3 (pp. 17564-17579). PMLR.

---

### Author Response · Authors · 2025-11-20

We thank all reviewers for the thoughtful and constructive comments. We are glad to hear that you found the contribution innovative and important. We followed your feedback closely and have carefully revised our paper as a result (see **updated PDF**). We further highlighted all key changes in **blue color**.

In particular, our **key improvements** are the following:

1. **New baseline**: We added new experiments with a new baseline. However, our proposed Survdiff performs best by a clear margin (see our **new results** in **Tables 2 and 3**).
2. **New empirical evaluations**: We added temporal event and censoring distribution plots, as well as normalized marginal covariate histograms to assess alignment between real and synthetic data, specifically real and synthetic time and event data. Across these diagnostics, our SurvDiff matches the real data most closely (see our **new Figures 4, 7-9** and our **new Appendix E**).
3. **New survival-specific fidelity metric**: We added the Event-Time Divergence to quantify survival-specific fidelity and to evaluate how well our model reproduces event-time distributions. SurvDiff achieves the best scores (see our **new Appendix C**).
4. **New Shape metric reporting**: We now also report the Shape metric to quantify overall distribution alignment. We added the Shape metric to quantify fidelity that includes survival-specific event-time. Our SurvDiff remains the strongest method on this metric (see our **new results** in **Table 2**).
5. **New ablation and parameter sensitivity studies**: We added an ablation study and a focused parameter sensitivity study to show how the key design choices influence performance. Overall, the studies confirm our design choice (see our **new Appendix J**).
6. **New privacy proof-of-concept and experiments**: We extended our method with a differentially private training variant that provides formal privacy guarantees. We also added a comparison to DP-GAN baseline and found that our differentially private variant achieves higher performance under the same privacy budget (see our **new Appendix I**).
7. **Clarifications of technical details**: We expanded implementation and evaluation details and added clarifications on definitions.

We believe these additions fully address the reviewers’ comments and strengthen both the empirical and methodological presentation. Thank you again, we sincerely appreciate your time and effort in reviewing our work and believe that the revised paper will be a strong fit for ICLR 2026.

---

### Public Comment · ~Marie_Brockschmidt1 · 2026-03-17

Dear reviewers, dear AC,

Thank you for the extensive feedback on our submission. We are glad to see so much encouraging feedback and, as a result, we have made a major revision of our paper. We agree that our initial submission had several weaknesses, and we made substantial changes (both post-review and post-rebuttal) to improve our work. Our latest version on arXiv addresses all of your points that came up during the review (including the points listed in the AC response).

Concretely, our **main improvements** are the following, which addressed all weaknesses of the initial submission:

1. **New baselines**: We added new experiments with new baselines. (see our **new results** in **Tables 2 and 3**).
2. **New empirical evaluations**: We added temporal event and censoring distribution plots, as well as normalized marginal covariate histograms to assess alignment between real and synthetic data  (see our **new Figures 4, 7-9** and our **new Appendix E**).
3. **New survival-specific fidelity metric**: We added a new metric called *Event-Time Divergence* to quantify how well our model reproduces event-time distributions (see our **new Appendix C**).
4. **New results for Shape metric**: We now also report the Shape metric to quantify overall distribution alignment *c* (see our **new results in Table 2**).
5. **New ablation and parameter sensitivity studies**: We added new ablation studies and a new parameter sensitivity study (see our **new Appendix J**).
6. **New privacy proof-of-concept and experiments**: We added an extension of our method that is differentially private and provides formal privacy guarantees (see our **new Appendix I**).

Importantly, all of the experiments in points (1)-(6) from above confirm our existing findings: **SurvDiff consistently shows strong performance.**

Thank you for the careful reviews. We are confident that, thanks to your feedback, our revised paper makes a strong submission to the community and is ready to be accepted at a future conference.

---

### Meta-Review · Area_Chair_zmjW · 2026-01-08

**Summary:**

This paper develops a diffusion model designed for survival data, which jointly generates covariates, event times, and censoring using a survival-specific loss.

- Missing/incorrect related work; novelty claim is questionable; mostly TabDiff + weighted Cox loss; not first diffusion for survival.

- Evaluation doesn’t match stated goals, and unclear whether downstream tasks test these.

- missing intuition/definitions (censoring, time-to-event, risk sets); hard notation in Sec 4.3.

- Missing implementation details for covariate distribution experiments

- Survival metrics weak; survival loss effectiveness questionable; baseline potentially weak

**Reviewer Concerns:**

Outstanding concern:

- Novelty is limited; soften/precisely scope “first diffusion model…” claims given prior diffusion-based survival setups

- Ablation specificity on weighting e.g. w=1 vs Eq. (11)

-  no explicit empirical privacy attacks for the default model. unclear privacy for non-DP model

- No new theoretical results

**Reviewer Scores:**

The reviewers were generally negative and the discussion might not have changed the overall sentiment much.

---

### Decision · Program_Chairs · 2026-01-26

Reject